# In situ mutational screening and CRISPR interference define *apterous* cis-regulatory inputs during compartment boundary formation

**Gustavo Aguilar\*†‡, Michèle E Sickmann†§, Dimitri Bieli#, Gordian Born¶, Markus Affolter, Martin Müller\***

Growth and Development, Biozentrum, Spitalstrasse 41, University of Basel, Basel, Switzerland

**\*For correspondence:**
gusag@mit.edu (GA);
m.mueller@unibas.ch (MM)

†These authors contributed equally to this work

**Present address:** ‡Department of Biological Engineering, Massachusetts Institute of Technology, Cambridge, United States; §Institute of Medical Virology, University of Zurich, Zurich, Switzerland; #Mabylon AG, Schlieren, Switzerland; ¶Arcondis, Reinach, Switzerland

## eLife Assessment

This **important** paper presents the discovery of the molecular basis of differential apterous expression during early *Drosophila* wing disc development. The evidence supporting these conclusions is **compelling**, ranging from classical genetic approaches to state-of-the-art genetic engineering techniques. By opening new questions, this paper is expected to be of broad interest to developmental biologists and geneticists working on transcriptional regulation.

**Abstract** The establishment of tissue axes is fundamental during embryonic development. In the *Drosophila* wing, the anterior/posterior (AP) and the dorsal/ventral (DV) compartment boundaries provide the basic coordinates around which the tissue develops. These boundaries arise as a result of two lineage decisions, the acquisition of posterior fate by the selector gene *engrailed* (*en*) and dorsal fate by the selector gene *apterous* (*ap*). While the *en* expression domain is set up during embryogenesis, *ap* expression begins only during early wing development. Thus, the correct establishment of the *ap* expression pattern relative to *en* must be tightly controlled. Here, we functionally investigate the transcriptional inputs integrated by the early *ap* enhancer (apE) and their requirement for correct boundary formation. Detailed mutational analyses using CRISPR/Cas revealed a role for apE in positioning the DV boundary relative to the AP boundary, with apE mutants often displaying mirror-image anterior wing duplications. We then designed and applied methods to accomplish tissue-specific enhancer disruption via dCas9 expression. This approach allowed us to dissect the spatiotemporal requirement for apE function, clarifying the mechanism by which apE misregulation leads to AP defects. Base-pair-resolution analyses of apE uncovered a single HOX-binding site essential for wing development that, when mutated, led to wingless flies. We demonstrated that the transcription factors Pointed (Pnt), Homothorax (Hth), and Grain (Grn) are required for apE function, and the HOX gene Antennapedia (Antp) contributes to early wing development. Together, our results provide a comprehensive molecular basis of early *ap* activation and the developmental consequences of its misregulation, shedding light on how compartmental boundaries are set up during development.

## Introduction

During development, cells segregate into different populations, often in a lineage-restricted manner. Between these populations, developmental boundaries arise as essential organizing centers,

providing the mechanical and chemical cues required for tissue patterning and growth (*Irvine and Rauskolb, 2001*). The *Drosophila* imaginal wing disc has served as a discovery platform for the study of developmental boundaries (*Tripathi and Irvine, 2022*). This tissue is subdivided by two lineage decisions. During embryogenesis, roughly half the cells of the future wing disc express the selector gene *engrailed* (*en*) (*Morata and Lawrence, 1975*). *en*-expressing cells acquire posterior (P) fate, while cells lacking *en* will adopt anterior (A) fate. These two populations will thereafter be segregated via the establishment of the anteroposterior (AP) compartment boundary (*Garcia-Bellido et al., 1973*; *Garcia-Bellido et al., 1976*). The AP boundary is essential for medio-lateral patterning of the wing, via the action of the morphogens Hedgehog (Hh) and Decapentaplegic (Dpp) (*Basler and Struhl, 1994*; *Capdevila and Guerrero, 1994*). The second lineage subdivision occurs during early wing disc development, when the gene *apterous* (*ap*) starts to be transcribed in a population of cells of the proximal wing disc (*Cohen et al., 1992*). *ap* encodes an LIM-homeodomain transcription factor (TF) responsible for specifying dorsal (D) identity, together with its cofactor CHIP (*Blair et al., 1994*; *Diaz-Benjumea and Cohen, 1993*; *Fernández-Fúnez et al., 1998*; *Garcia-Bellido et al., 1976*; *Milán and Cohen, 1999*; *van Meyel et al., 1999*). Cells lacking *ap* expression become ventral (V). The DV boundary instructs proximo-distal patterning via the morphogen Wingless (Wg) (*Couso et al., 1993*; *Neumann and Cohen, 1997*; *Zecca et al., 1996*). At the intersection of these boundaries, the secreted signaling molecules expressed at the DV (Wg) and the AP (Dpp) boundaries synergistically promote specification and growth of the wing pouch, the larval precursor of the wing blade (*Couso et al., 1995*; *Kim et al., 1997*; *Williams et al., 1994*). The two boundaries thereby provide the coordinate system according to which the four segregated quadrants of the developing wing (anterior-dorsal, AD; anterior-ventral, AV; posterior-dorsal, PD; and posterior-ventral, PV) develop.

The downstream effectors of *ap* during wing development are well characterized (*Blair et al., 1994*; *Klein et al., 1998*). However, little is known about how the expression pattern of *ap* is set up with respect to that of *en*. In particular, the mechanisms that ensure proper spatial alignment between the DV and AP boundaries remain incompletely understood.

Interestingly, *ap^blot^*, a mutation within the *ap* cis-regulatory landscape, has been shown to produce mirror-image transformations of the P compartment into the A compartment (*Whittle, 1979*). How such interaction between *ap* and the AP specification program arises is unknown. Previously, we have found that *ap^blot^* maps to the *ap* Early enhancer (apE), the regulatory module responsible for *ap* early transcription (*Bieli et al., 2015a*). In later stages, *ap* positively regulates itself via the enhancer apDV, at least in the wing pouch (*Figure 1—figure supplement 1*, *Bieli et al., 2015b*). This regulatory feedback is likely to maintain the expression pattern established earlier by apE. The main upstream factor known to regulate apE is the EGFR pathway. EGFR signaling has been shown to be both required and sufficient for *ap* transcription (*Wang et al., 2000*; *Zecca and Struhl, 2002*).

In this study, we have taken a bottom-up approach, from the sequence of the enhancer to the development of the tissue. We have genetically dissected apE using CRISPR gene editing and found that apE is required for correct DV positioning. Our results provide mechanistic insight into how perturbations of apE impact DV/AP boundary organization and wing patterning. apE hypomorphic mutants displayed problems in the growth of the posterior-dorsal compartment and subsequent mirror-image duplications of the A compartment. The EGFR core TF Pointed (Pnt) and the TF Homothorax (Hth) are responsible for early *ap* expression via apE. The loss of these factors in the posterior compartment resulted in mirror-image duplications of the wing. Base-pair-resolution mutation analyses of apE subregions uncovered a GATA–HOX complex fundamental for wing formation, of which we identify Grain (Grn) as the GATA TF and propose Antennapedia (Antp) as a candidate HOX input. Finally, in order to dissect the spatiotemporal requirement of apE, we have implemented a novel methodology for enhancer inactivation, solely based on the recruitment of catalytic-dead Cas9 (dCas9) to the enhancer. Using this tool, we validated the results of our mutational analyses and dissected the spatial requirement of apE function.

## Results
### Precise manipulation of the endogenous apE enhancer
The 27 kb intergenic spacer upstream of the *ap* transcription start site (TSS) contains five conserved regions, named C1–C5 (*Bieli et al., 2015b*). We have previously replaced this entire intergenic

sequence by an *attP* site, thereby generating a 'landing site' in the locus. Subsequently, different attB-containing fragments were re-integrated via φC31-mediated transgenesis. This approach revealed three distinct modules required during wing development: the apE enhancer, within C2, required for early *ap* expression; the DV enhancer within C5, responsible for robust *ap* expression along the DV boundary; and a Polycomb response element (PRE), immediately upstream to the TSS, required for maintenance of the transcriptional state of *ap* (**Bieli et al., 2015b**). While useful to study the overall cis-regulatory architecture, re-integration into this landing site resulted in changes in the relative distance of these individual regulatory modules with respect to the TSS and entailed large genomic deletions. In order to study apE in its endogenous context, we decided to construct a new landing site.

The minimal enhancer region contained in apE encompasses 463 bp in the proximal area of C2, hereafter referred to as OR463 (**Figure 1—figure supplement 2**). We have previously shown that re-integration of OR463 alone is sufficient to rescue the wing phenotype produced by the loss of C2 (**Bieli et al., 2015b**). Based on these results, we decided to substitute OR463 with an *attP* site using CRISPR/Cas9 editing (**Figure 1—figure supplement 3** and Materials and methods). We have also demonstrated that crosstalk between apE and *dad* enhancers is responsible for the dominant *Xasta* (*Xa*) phenotype (**Bieli et al., 2015a**). Thus, in order to positively select for CRISPR/Cas9-mediated HDR, the Dad13 enhancer was inserted, flanked by FRTs, alongside the *attP* site (**Figure 1—figure supplement 3A, B**). As predicted, heterozygous insertion events of the *attP* coupled to Dad13 were easily identified by their characteristic wing phenotypes (**Figure 1—figure supplement 3A'**). Flippase was subsequently used to excise Dad13 (**Figure 1—figure supplement 3B**). The resulting substitution of OR463 by the *attP* site resulted in fully penetrant loss of wings in homozygotes (**Figure 1F**, **Figure 1—figure supplement 3B**). The *attP* site was then used to re-integrate the different OR463 fragments via φC31-mediated insertion, along with the *mini-yellow* marker, which was later removed using the FRT/Flippase system (**Figure 1—figure supplement 3C, D**).

Re-integration of OR463 into this landing site resulted in 100% normal wings in homozygotes (**Figure 1—figure supplement 3D'**) and 95% normal wings in hemizygotes (over the $ap^{DG3}$ deletion, which encompasses the whole intergenic spacer **Bieli et al., 2015b**). This approach was used to generate all the apE mutants described in this study.

## OR463 contains two highly conserved sequences required for wing development

Bioinformatic analysis of OR463 uncovered four main modules based on sequence conservation among related insects (**Figure 1A**, Materials and methods). We further refined those modules by analyzing the conservation of TF-binding sites using MotEvo (**Arnold et al., 2012**). This analysis allowed us to subdivide OR463 into four highly conserved regions (m1–m4) and six less well-conserved inter-regions (N1–N6) (**Figure 1B**, **Supplementary file 1**).

To evaluate the functional requirement of each region, we generated fly strains bearing precise deletions of each of them. Homozygous and hemizygous animals for all the mutants reached adulthood, displaying altered mutant phenotypes only within the wings and halteres. Scoring of wing phenotypes uncovered two genomic regions which, when deleted, resulted in a higher penetrance of mutant phenotypes (**Figure 1C**). The first area was centered in m3, whose deletion resulted in fully penetrant loss of wings (**Figure 1G**). Deletion of the neighboring region N4 also yielded a very high number of defective wings. The second sensitive area was located around conserved region m1 (N1, m1, N2), with both *Δm1* and *ΔN2* deletions yielding similarly penetrant phenotypes. Phenotypic penetrance was further increased when the aforementioned mutant alleles were scored in a hemizygous background (over $ap^{DG3}$) (**Figure 2D**).

Excluding the *Δm3* deletion, the other mutants displayed similar phenotypes, although varying in quantity and quality. Interestingly, many of these wings presented a posterior compartment outgrowth, in which the vein pattern was reminiscent of that of the anterior wing, and anterior margin bristles were found in its posterior edge (**Figure 1H**). In those wings, the anterior compartment was largely unaffected.

In some of the most penetrant alleles, and especially when the mutants were in a hemizygous condition, wings did not present an outgrowth, but rather partial mirror-image transformations of the margin bristles of the P compartment into A compartment identity, accompanied by unusual vein trajectories in the P compartment (**Figure 1I**). This wing phenotype was often associated with

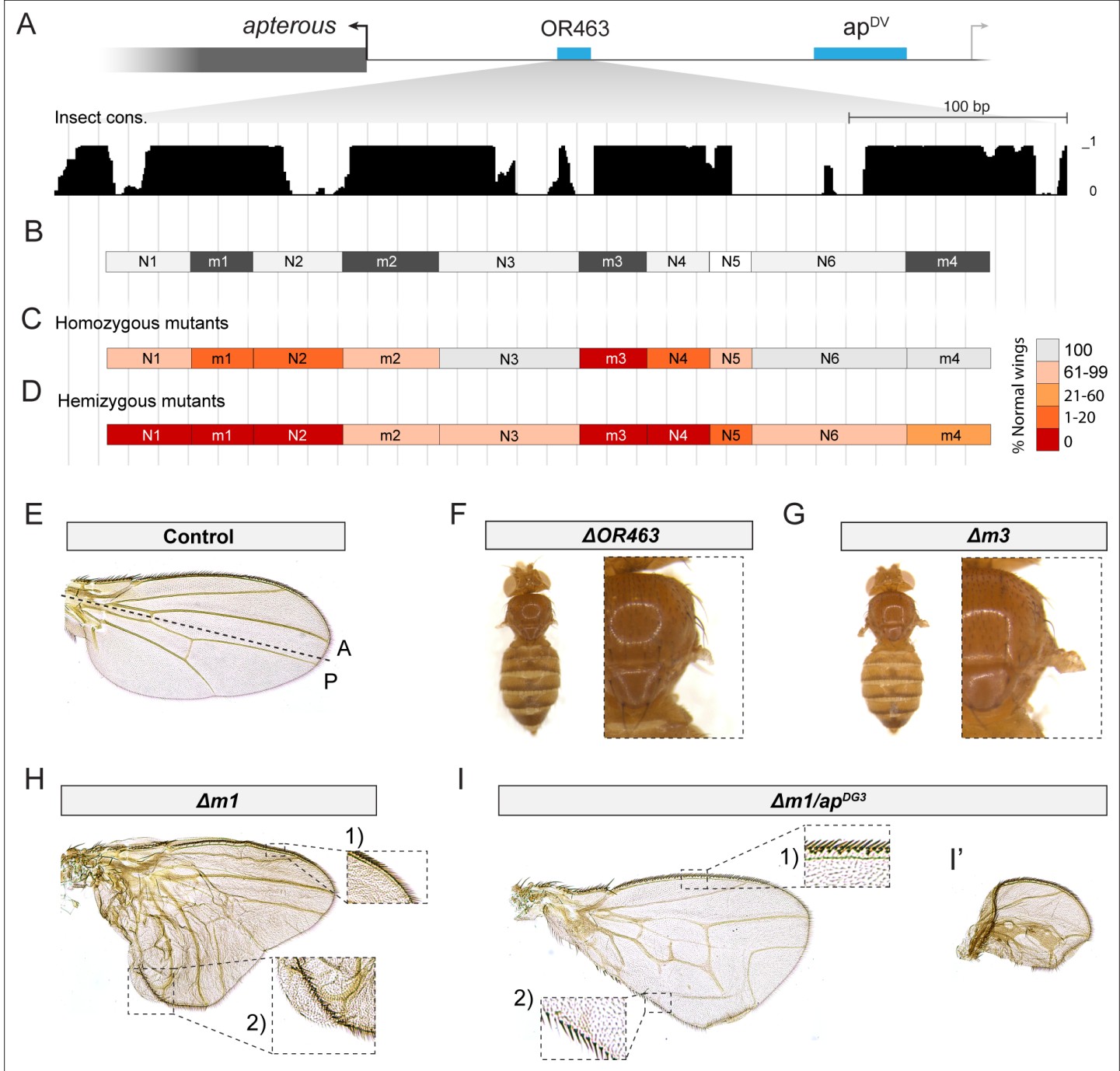

**Figure 1.** Two conserved regions within OR463 are fundamental for wing development. (**A**) Illustration of *ap* upstream intergenic region. In blue: enhancer sequences relevant during wing development (*Bieli et al., 2015b*). In black: OR463 sequence conservation among related insects. (**B**) Different regions in which OR463 was divided for mutation analysis. In dark gray, sequences with highly conserved transcription factor (TF)-binding sites. (**C**) Percentage of wild-type wings when each of the fragments was deleted in homozygotes and (**D**) hemizygotes (over the $ap^{DG3}$ allele). (**E**) Control wing (Re-integrated WT OR463 in $ap^{R2}$ landing site). (**F**) Loss of wing displayed by *ΔOR463* mutants. (**G**) Loss of wing displayed by *Δm3* mutants. (**H**) Representative wing phenotype derived from *Δm1* deletion in homozygotes. The A compartment presents correct venation, while the P compartment presents an outgrowth. Venation pattern in this outgrowth is disturbed. Inset 1: Detail of the A compartment bristles. Inset 2: Bristles of A identity in the P compartment. (**I**) Representative wing phenotypes derived from *Δm1* deletion in a hemizygous background. P compartment venation pattern is disturbed. No outgrowth is formed but the P compartment presents severe venation defects, with some veins positioned perpendicularly to its normal direction. Inset 1: Detail of A bristles. Inset 2: Detail of A bristles in the P compartment. Note that transformation of the P margin into A is not complete. (**I'**) Wing phenotype also present in *Δm1* hemizygous flies. In these, the P compartment is severely reduced and mostly A bristles are present in the margin.

*Figure 1 continued on next page*

*Figure 1 continued*

The online version of this article includes the following figure supplement(s) for figure 1:

**Figure supplement 1.** C2 and C5 activity throughout larval development.

**Figure supplement 2.** Identification of the minimal apE enhancer within apC2.

**Figure supplement 3.** Generation and validation of the OR463 landing site.

a reduction in P compartment size. In some of these cases, the P compartment was hardly detectable, and the overall wing size was severely reduced, resulting in roundish wings in which only A and tip margin bristles were present (*Figure 1I'*). Some of these phenotypes are reminiscent of those reported for *ap^Blot^* (*Whittle, 1979*) and point toward a previously undescribed crosstalk between *ap* early expression and the AP specification program.

## OR463 disruption results in misplacement of the DV boundary with respect to the AP boundary

In order to dissect how these mirror-image duplications arise, we analyzed the expression pattern of Ap in third-larval stage (L3) wing imaginal discs. In control discs (wild-type OR463 re-integrated into the landing site), the Ap expression domain (*Figure 2A*) mimicked that seen in wild-type discs (*Bieli et al., 2015b*). Wg was detected in its characteristic stripe along the DV boundary and in the two concentric rings of the hinge (*Figure 2A*). In Δm1 homozygous mutants, disruption of the Ap expression pattern was most evident in the P compartment, where the DV boundary was curved toward the hinge (*Figure 2A'*). This deformation of the DV boundary was also revealed by Wg immunostaining (*Figure 2A'*). In these cases, Wg expression in the P hinge also seemed altered. The deformation of DV compartmentalization was also observed in other alleles carrying deletions within the OR463 region, as shown by anti-Ap immunostaining (*Figure 2—figure supplement 1*). As described for adult wings, the severity of the wing disc phenotypes was exacerbated in a hemizygous background, with wing discs that presented a more obvious deformation of the Ap pattern in the P side of the wing and overall reduced size (*Figure 2A''*). In all cases, the intensity of anti-Ap staining was comparable between the wild-type and the mutant discs, suggesting a role of apE in setting up the proper *ap* expression domain rather than its expression level.

P to A mirror-image duplication of the wing has been described upon different genetic perturbations (*Baonza et al., 2000*; *Brower, 1984*; *Guillén et al., 1995*; *Hidalgo, 1998*; *Recasens-Alvarez et al., 2017*; *Uemura et al., 1993*). A paradigmatic case is the loss of *en* expression. *en* mutant clones in the P compartment lead to loss of P identity and the formation of a new AP boundary around the clone (*Brower, 1984*; *Tabata et al., 1995*). Thus, we analyzed En expression in wing discs of Δap^E hemizygous mutants (*Figure 2—figure supplement 2*). While wing disc size is severely affected compared to control, En localization, as revealed by immunostaining, presented little change in its expression pattern. To account for the possibility that *en* expression would be lost in some cells during development, thereby leading to the duplications, we analyzed the lineage of *en* positive cells in both control and Δm1 hemizygous background. To that end, we made use of gTRACE (*Evans et al., 2009*), a genetic tool that uses the expression of a Gal4 line of interest to both immortalize a fluorescent reporter (via Flp-mediated excision of a FRT cassette, leading to the constitutive expression of a *tub*-GFP reporter) and at the same time reveal current Gal4 levels (via an *UAS*-RFP transgene). In both control and Δm1 hemizygous background, *en*-Gal4 revealed highly overlapping patterns of lineage and current expression (*Figure 2—figure supplement 2B, C*). Thus, these experiments rule out loss of *en* expression as the driver for the duplications observed in *ap* mutants. In addition, the experiment revealed that in Δm1 mutants, the P compartment is severely reduced and that the position of the AP boundary along the pouch was also distorted (*Figure 2—figure supplement 2C*).

Motivated by this finding, we measured the relative size of the P wing disc compartment in several of the OR463 mutants and observed a reduction in all cases (*Figure 2B*), in agreement with the results of the gTRACE experiment and the phenotypes of adult wings (*Figure 1G*). The relative position of the DV and AP boundaries was then studied by co-immunostaining of Ap and Ptc proteins. In control discs, DV and AP boundaries form perpendicular axes, with a single intersection in the middle of the wing pouch, dividing the disc in four quadrants (DA, DP, VA and VP; see *Figure 2C*). In Δm1 hemizygous discs, PD quadrant size was strongly reduced (*Figure 2C'*). This resulted in two intersection

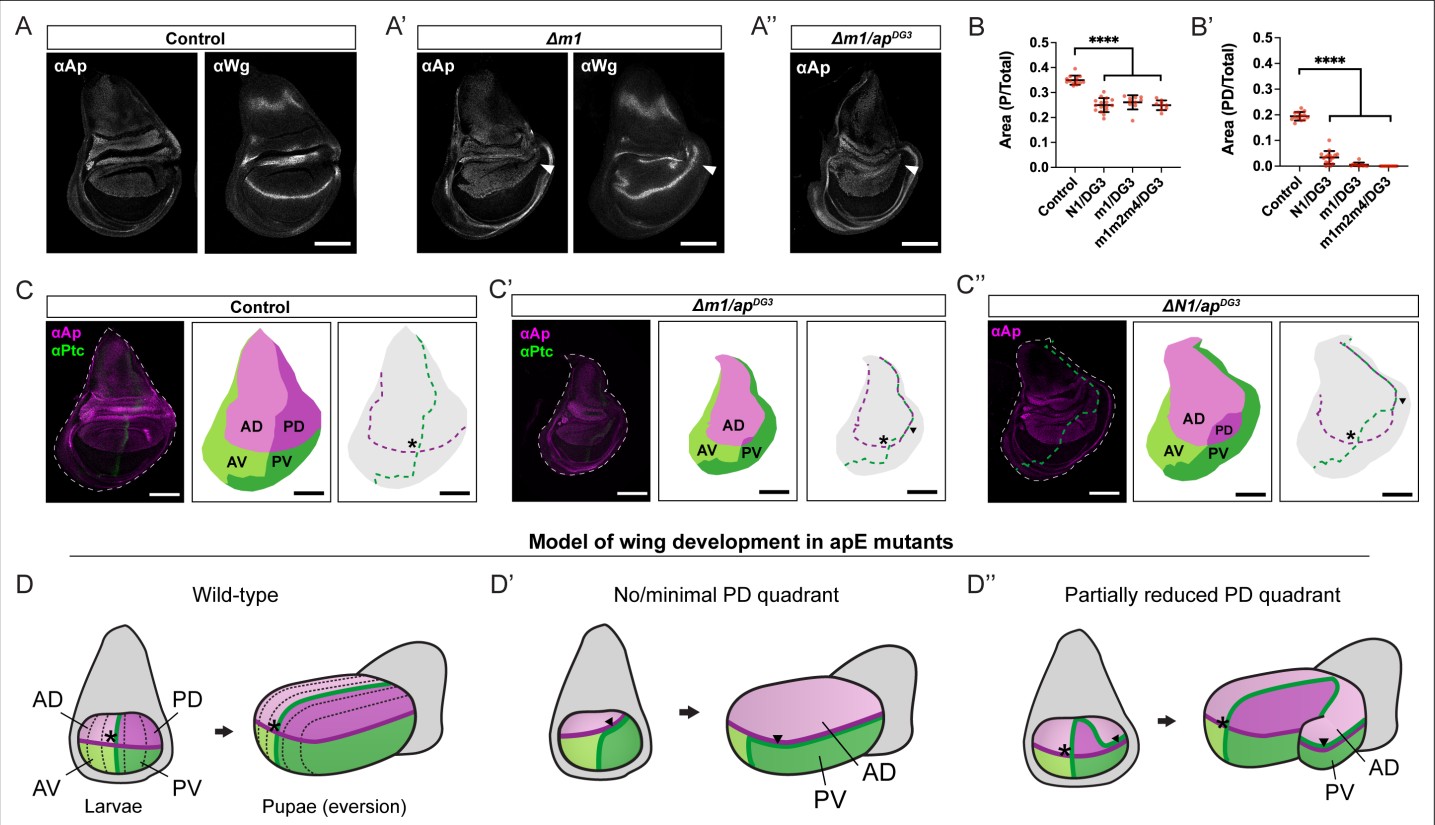

**Figure 2.** Mirror-image duplications arise due to changes in DV and AP boundary position. (**A**) Anti-Ap and anti-Wg immunostaining of control wing discs (Re-integrated WT OR463). (**A'**) Anti-Ap and anti-Wg staining of wing discs in homozygous Δm1 mutants. The DV boundary is distorted in the P compartment, where it is extended into the presumptive wing hinge (arrowheads). (**A"**) Anti-Ap staining of Δm1 hemizygous wing discs. The DV boundary is further deformed in the P compartment (arrowhead). (**B**) Quantification of the relative P size (P Area/Total Area) and relative PD size (PD Area/Total Area) in different mutants (p-value <0.0005, control: $n$ = 13, N1/DG3: $n$ = 16, Δm1/DG3: $n$ = 10, Δm1m2m4/DG3: $n$ = 9). (**C, C', C"**) Relative position of AP and DV boundaries as revealed by immunostaining with anti-Ap and anti-Ptc in wing discs of control, Δm1/DG3, and ΔN1/DG3. Different quadrant maps are then subtracted and the AP and DV boundaries represented with dashed lines (red and blue, respectively). Asterisks depict the intersections of AP and DV boundaries. Scale bars: 100 μm. (**D**) Schematic of wild-type wing development from larval to pupal stages. In the larval disc, the AP boundary (green) intersects the DV boundary (purple) at a single point (asterisk). During pupal eversion, the wing pouch unfolds along the DV boundary, which becomes the future wing margin, and veins are patterned in parallel to the AP boundary. (**D'**) Model in which PD growth is strongly reduced or nearly absent. After eversion, the anterior compartment will present anterior identity on its dorsal side and posterior identity on its ventral side. This mixed orientation accounts for the partial A identity observed at the posterior border. (**D"**) Model in which PD size is reduced but still occupies a considerable pouch area. Here, the AP and DV boundaries meet at two distinct positions within the pouch, spaced far enough apart to support the specification of two wing organizing centers. The arrow heads mark the region where AP and DV boundaries overlap within the posterior compartment. After eversion, the secondary center generates a wing structure with dorsal A identity and ventral P identity, producing characteristic venation defects patterned along the ectopic DV boundary.

The online version of this article includes the following source data and figure supplement(s) for figure 2:

**Source data 1.** Quantification of relative P and PD sizes in different mutants.

**Figure supplement 1.** Ap localization in various genotypes.

**Figure supplement 2.** Loss of *en* expression cannot explain the mirror-image duplications of apE mutants.

**Figure supplement 3.** Vestigial expression in OR463 mutants.

**Figure supplement 4.** Model for the development of mirror-image duplications in ap OR463 mutants.

**Figure supplement 5.** Posterior hinge and Upd expression patterns are affected in *ap* mutants.

points and long stretches where both the DV and the AP coincide. In cases where the size of PD was less compromised, as in Δ*N1* hemizygous mutants (***Figure 2C"***), these two intersections were located further apart. Quantification of PD quadrant size revealed a great decrease of relative area in different OR463 mutants (***Figure 2B'***).

To identify the nature of the posterior outgrowths, we performed anti-Vestigal (Vg) antibody staining of *Δm1* mutants (*Figure 2—figure supplement 3A*). Vg is a key regulator of wing specification and also participates in wing growth and patterning (*Baena-Lopez and García-Bellido, 2006*; *Kim et al., 1996*; *Zecca and Struhl, 2007a*). In those discs, in which the stripe was extended and the P compartment was enlarged, Vg was detected throughout the outgrowth, supporting the wing pouch identity of this region (*Figure 2—figure supplement 3B*). Hemizygous *Δm3* mutants presented a highly reduced anti-Vg signal, which suggests that no wing pouch is specified in these mutants (*Figure 2—figure supplement 3C*).

## A model to explain the adult patterning defects of OR463 mutants

Intersection of the AP and DV boundaries is crucial for wing specification, patterning and growth via Dpp and Wg signals. Wild-type wings present a single intersection of the AP and DV boundaries, around which the four wing quadrants (DA, DP, VA, and VP) grow coordinately. During morphogenesis, the disc evaginates along the DV boundary to form the adult wing (*Aldaz et al., 2010*; *Fristrom and Fristrom, 1975*; *Figure 2D*, *Figure 2—figure supplement 1A*). Our results indicate that OR463 is fundamental for setting up the *ap* expression pattern, and that the P compartment is the most sensitive to its mutation. We therefore sought to explain the diversity of adult phenotypes by considering how changes in the ap expression domain alter the geometry of the P compartment. We hypothesized that PD size operates as a threshold parameter, such that the number of DV/AP boundary intersections, as well as the distance between them, explains the distinct adult phenotypes observed across OR463 mutants.

1. In the cases when PD is very small or absent, no outgrowth would be observed, as there is only one intersection point, or two intersections that are very close to each other. In these cases, both AP and DV boundaries will largely coincide (*Figure 2D'*, *Figure 2—figure supplement 4C*). Upon eversion, the PV compartment will be opposed by tissue with AD identity, thus resulting in partial transformation of the margin and venation patterns into A identity. This is in accordance with the results presented in *Figure 1*, where most of the adult wings analyzed presented only partial transformation of the wing margin. This scenario also explains why the venation pattern is often disrupted, since during apposition, dorsal and ventral epithelia will present different venation patterns.
2. When PD is reduced but still occupies a significant area (as in N1/DG3 animals, see *Figure 2C*), the wing disc presents two intersections that are located far from each other. In these cases, two wing pouches would be specified and an outgrowth would be formed (*Figure 2D''*, *Figure 2—figure supplement 4B*). In the wing margin of this outgrowth, both AP and DV boundaries coincide. We predict that the D side will present A fate in this outgrowth, while the V side will have P fate, resulting in an anterior mirror-image duplication of dorsal structures.
3. Finally, in the cases in which *ap* expression is depleted from P and affected in the A compartment, AP and DV boundaries do not intersect. However, if the boundaries are close enough, they would induce slight wing pouch growth (*Figure 2—figure supplement 4D*). In these wing discs, the DV boundary will be fully embedded in the A compartment, and thus, all the wing margin will present A identity. Consistent with this prediction, the wings showing the most drastic size defects among the different OR463 mutants only presented A and tip margin (*Figure 1I'*).

This model predicts that in both the P outgrowths (*Figure 2—figure supplement 4B*) and the P to A mirror-image duplications (*Figure 2—figure supplement 4C*), the D compartment presents AD identity. Consistent with our model, we could identify campaniform sensillae in *Δm1* homozygous mutant wings (*Figure 2—figure supplement 4E*) as well as in additional OR463 mutant backgrounds (data not shown); these sensillae are specialized sensory organ only present in the AD compartment of the wing blade (*Dinges et al., 2021*). Together, these observations support a PD-size threshold model that explains how the diverse adult phenotypes arise in OR463 mutants.

## The posterior hinge is severely affected in OR463 mutants

The wing pouch grows surrounded by (and at the expense of *Zecca and Struhl, 2007b*) the presumptive wing hinge. The hinge does not have a passive role in wing disc growth but also provides growth factors required for cell division. In particular, JAK/STAT ligands (Unpaired 1–3) derived from the hinge have been found to promote posterior growth (*Recasens-Alvarez et al., 2017*). Loss of JAK/STAT signaling in P cells results in mirror-image duplications that highly resemble those seen in OR463

mutants. *ap* expression is indispensable for hinge development (*Metz, 1914*). Indeed, apE mutants totally lack this structure (*Bieli et al., 2015b*). Thus, we analyzed the morphology of the hinge in OR463 mutants. We labeled the hinge by immunodetection of the hinge-specific TF Homothorax (Hth). Compared to control discs, in which the P proximal hinge is present (*Figure 2—figure supplement 5A*), hemizygous Δm1 mutants completely lacked this region (*Figure 2—figure supplement 5A'*).

We then assessed the expression of the JAK/STAT ligand Unpaired 1 (Upd1). *upd-1* is expressed in five areas in third instar wing imaginal discs, in three main spots in the dorsal hinge and in two weaker regions in the ventral body wall (*Johnstone et al., 2013*). This pattern was faithfully captured via an *upd*-Gal4 line driving the expression of *UAS*-CD8:GFP (*Figure 2—figure supplement 5B*). As expected, given the strong hinge phenotypes of OR463 mutants, *upd*-Gal4>*UAS*-CD8:GFP did not display any P signal in the proximal area of the wing when OR463 was deleted (*Figure 2—figure supplement 5B'*).

These results demonstrate that the main P source of JAK/STAT signaling in the disc is absent in OR463 mutants, suggesting that the similar phenotypes produced by defective JAK/STAT signaling and by mutations in OR463 might be related.

## Spatiotemporal analysis of apE via dCas9

The functional dissection of regulatory regions has classically relied on the generation of mutants. Such mutant analyses, however, do not provide spatiotemporal resolution of enhancer function. Instead, reporters are normally employed as indirect readouts. Recently, expression of catalytically dead Cas9 (dCas9) has been proposed as a means to inhibit TF-DNA binding in cell culture (*Shariati et al., 2019*) within a CRISPR interference (CRISPRi) framework (*Qi et al., 2013*). Given the high sensitivity of OR463 to perturbation, we hypothesized that localized expression of dCas9 could cause steric hindrance, and thereby block enhancer activity (*Figure 3A*).

To achieve spatiotemporal control over dCas9 expression, we constructed a fly line in which dCas9 was placed under the control of *UAS* enhancer sequences. Expression of dCas9 by the posterior driver *en*-Gal4 did not result in major phenotypes at 23 or 25°C when combined with a control gRNA (*Figure 3C*, see Materials and methods), indicating that dCas9 does not overtly perturb cellular physiology. To target dCas9 to the OR463 locus, we generated a transgene encoding four ubiquitously expressed gRNAs (*U6*-OR463.gRNAx4). Three of these gRNAs targeted the vicinity of m3 (*Figure 3B*), the region that exhibited the strongest phenotypes in our deletion analysis (*Figure 1*).

Our mutation analysis indicated that loss of OR463 activity in the P compartment is responsible for the mirror-image duplications observed in OR463 mutants (*Figure 2*, *Figure 2—figure supplement 3*). We directly tested this assumption by expressing dCas9 using *en*-Gal4 in the presence of OR463 gRNAs. This manipulation produced wings with posterior defects (*Figure 3D*) that closely resembled those observed in the genetic deletions (*Figure 1*). Interestingly, the phenotype most often detected at 23°C was the presence of P outgrowths (with some A bristles). Increasing dCas9 expression (at 25°C) yielded a majority of wings in which the posterior edge was partially transformed into A identity. The wing discs obtained from these experiments (at 23°C) displayed a reduced PD quadrant together with mispositioning of the DV and AP boundaries, which intersected twice (*Figure 3E, F*). These experiments independently confirm the crucial role of OR463 in correctly positioning the DV boundary and demonstrate that **posterior loss of apE activity** drives the AP mirror-image duplications. To confirm the spatial specificity of enhancer inhibition, dCas9 was also expressed using *ptc*-Gal4 in the presence of OR463 gRNAs. In this case, the *ap* expression domain was divided in two, with no expression along the central pouch (*Figure 3G*). This broad repression band coincides with the expression pattern of *ptc* in early wing discs, demonstrating that dCas9 can inhibit enhancer function with high spatial precision in vivo.

To define the developmental time window during which the apE enhancer remains sensitive to repression, we combined the temperature-sensitive *tub*-Gal80$^{ts}$ system with temporally controlled expression of dCas9. Animals carrying the *en*-Gal4, *tub*-Gal80$^{ts}$, *UAS*-dCas9 and *U6*-OR463.gRNAx4 transgenes were maintained at 18°C to suppress dCas9 expression. Independent sets of embryos were then shifted to 29°C at successive developmental intervals ranging from 0 to 168 hr after egg laying (AEL), so that dCas9 induction occurred at distinct time points in development (*Figure 3H*). Under these conditions, dCas9 transcription was induced only after the temperature shift, while the

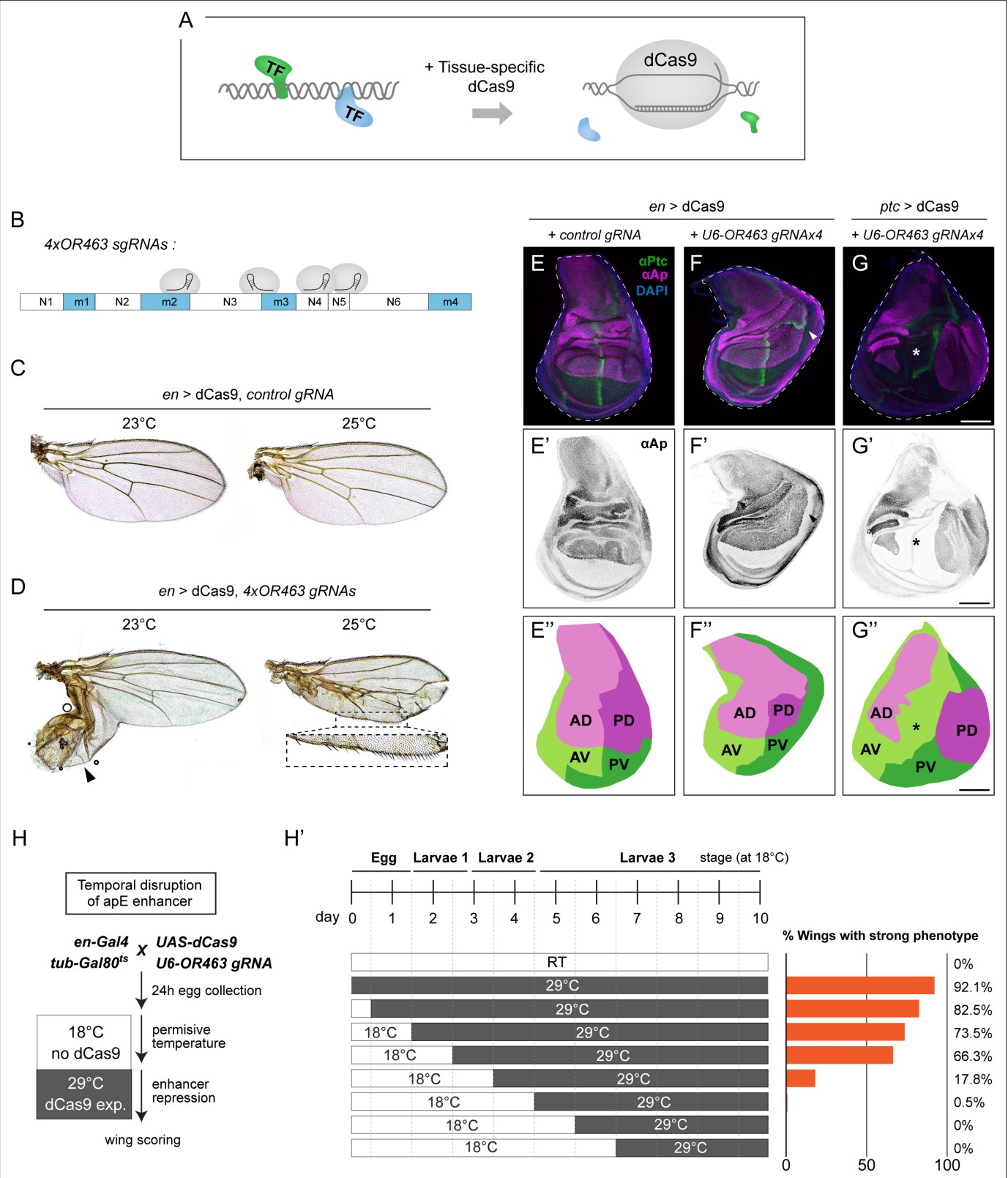

**Figure 3.** Spatiotemporal characterization of OR463 via localized dCas9 expression. (**A**) Schematic representation of the method. Upon localized expression, dCas9 would bind to DNA displacing or interfering with transcription factor (TF) binding. (**B**) Target sites of the gRNAs present in the *U6-OR463.gRNAx4* transgene. (**C**) Wings of animals expressing dCas9 under the *en*-Gal4 driver in the presence of the control gRNA at 23 and 25°C. (**D**) Wings of animals expressing dCas9 under the *en*-Gal4 driver in the presence of *U6*-OR463.gRNAx4 at 23 and 25°C. Arrowhead indicates the P

*Figure 3 continued on next page*

*Figure 3 continued*

outgrowth. Notice the presence of A bristles in the P edge (dashed box). (**E**) Control wing discs, expressing dCas9 under the *en*-Gal4 driver in the presence of control gRNAs at 23°C. (**E'**) Anti-Ap immunostaining showing the normal Ap expression pattern in late wing discs. (**E''**) Quadrant map subtracted from panel E. Notice the single perpendicular intersection between AP and DV boundaries. (**F**) Anti-Ptc and anti-Ap immunostaining of wing discs expressing dCas9 under the *en*-Gal4 driver in the presence of *U6*-OR463.gRNAx4 at 23°C. Notice the P outgrowth (arrowhead). (**F'**) Anti-Ap immunostaining exhibiting the extended posterior pattern in the posterior compartment. (**F''**) Quadrant map subtracted from panel F. Notice the reduced size of PD quadrant. AP and DV contact twice, once perpendicularly as in control discs and once tangentially in the P outgrowth. (**G**) Anti-Ptc and anti-Ap immunostaining of wing discs expressing dCas9 under the *ptc*-Gal4 driver in the presence of *U6*-OR463.gRNAx4 at 23°C. Notice lack of Ap signal along the central wing disc (asterisk). (**G'**) Anti-Ap immunostaining of the disc in G. (**G''**) Quadrant map subtracted from panel G. Notice the lack of contact between AD and PD quadrants. Scale bars: 100 µm. (**H**) Schematic of the temperature-shift experiment used to define the developmental window during which the apE enhancer is sensitive to repression. *en*-Gal4, *tub*-Gal80ᵗˢ, UAS-dCas9; *U6*-OR463.gRNA(4×) animals were kept at 18°C to suppress dCas9 and shifted to 29°C at successive developmental intervals (0–168 hr after egg laying [AEL]) to induce dCas9 expression at distinct stages. (**H'**) Timeline summarizing the temperature regime and the percentage of adult wings showing strong apterous-like phenotypes following induction at each time point. Early induction (0–48 hr AEL) produced the highest penetrance, with progressively weaker effects after 72 hr AEL.

gRNAs were expressed constitutively. Wing phenotypes were quantified in adult progeny as a readout of apE enhancer perturbation. When dCas9 was expressed from embryonic or early larval stages (0–48 hr AEL), nearly all wings (70–90%) displayed severe *ap*-like phenotypes, including posterior compartment duplication and loss of anterior–posterior boundary integrity. Shifting animals later (48–72 hr AEL) still produced a majority (~66%) of abnormal wings, whereas induction after 72 hr AEL resulted in progressively weaker effects and complete loss of phenotypes by 96 hr AEL (*Figure 3H'*).

These results define the developmental window during which apE activity is required for proper wing patterning. Perturbation during the first half of the second larval instar (≤96 hr at 18°C) was sufficient to elicit strong *ap*-like transformations, consistent with the enhancer being functionally required during early larval stages and becoming dispensable thereafter. The temporal decline in phenotype penetrance thus reflects the progressive loss of apE sensitivity to dCas9-mediated repression, pinpointing when its activity is no longer required for wing morphogenesis.

## Pnt and Hth are required for m1–m4 activity

Having established the requirement of OR463 activity during wing development and boundary positioning, we analyzed the role of TF potentially responsible for its activity pattern. m1 bioinformatic analyses identified two main TF-binding sites, predicted to be recognized by Hth and a TF from the E26 transformation-specific (ETS) family (*Figure 4A*). The m4 region was also predicted to be recognized by ETS TFs, however, its deletion had minor effect (*Figure 1C*). In order to accentuate the possible phenotypes caused by deletions within apE, all subsequent m1 deletions were generated in a background which lacked m4. We generated small deletions encompassing each of the putative binding sites (*Figure 4A*). Scoring of the phenotypic penetrance in homozygotes (*Figure 4B*) and hemizygotes (*Figure 4B'*) revealed that both sites were important for wing development, the ETS-binding site having the biggest effect when deleted. Analysis of DV and AP boundary position based on immunostaining revealed that deletion of the different binding sites in a hemizygous background resulted in a reduced size of the PD quadrant (*Figure 4C*). Adult wings of the different mutants resembled the loss of m1, amid a reduction in the penetrance of the phenotypes (data not shown).

We then knocked-down the expression of the TFs predicted to bind m1, Pnt and Hth, via RNAi expression in the posterior compartment using *en*-Gal4. Affected cells were labeled using *UAS*-CD8:GFP. Strikingly, when compared to control wing discs, *hth*RNAi expression resulted in drastic reduction of the PD quadrant, mimicking the loss of m1 (*Figure 4D*). Unfortunately, adult phenotypes could not be studied, since these animals did not reach adulthood.

We have previously demonstrated that the ETS factor Pnt is able to bind apE (*Bieli et al., 2015b*), making it an ideal putative m1 input. In addition, deletion of most conserved Pnt-binding sites in an apE reporter reduced its activity (*Bieli et al., 2015b*). Expression of *pnt*RNAi resulted in a reduction of the PD quadrant (*Figure 4D*). In this case, a posterior outgrowth was formed, mimicking that of the phenotypes caused by the loss of the ETS-binding site in m1. *en*Gal4>*pnt*RNAi animals reached adulthood, displaying outgrowths in the P compartment (*Figure 4E*), confirming previous reports (*Baonza et al., 2000*). As it is the case for some of the different OR463 mutants, these outgrowths presented partial P to A transformation. In addition, campaniform sensillae were identified in the transformed compartment (*Figure 4E*).

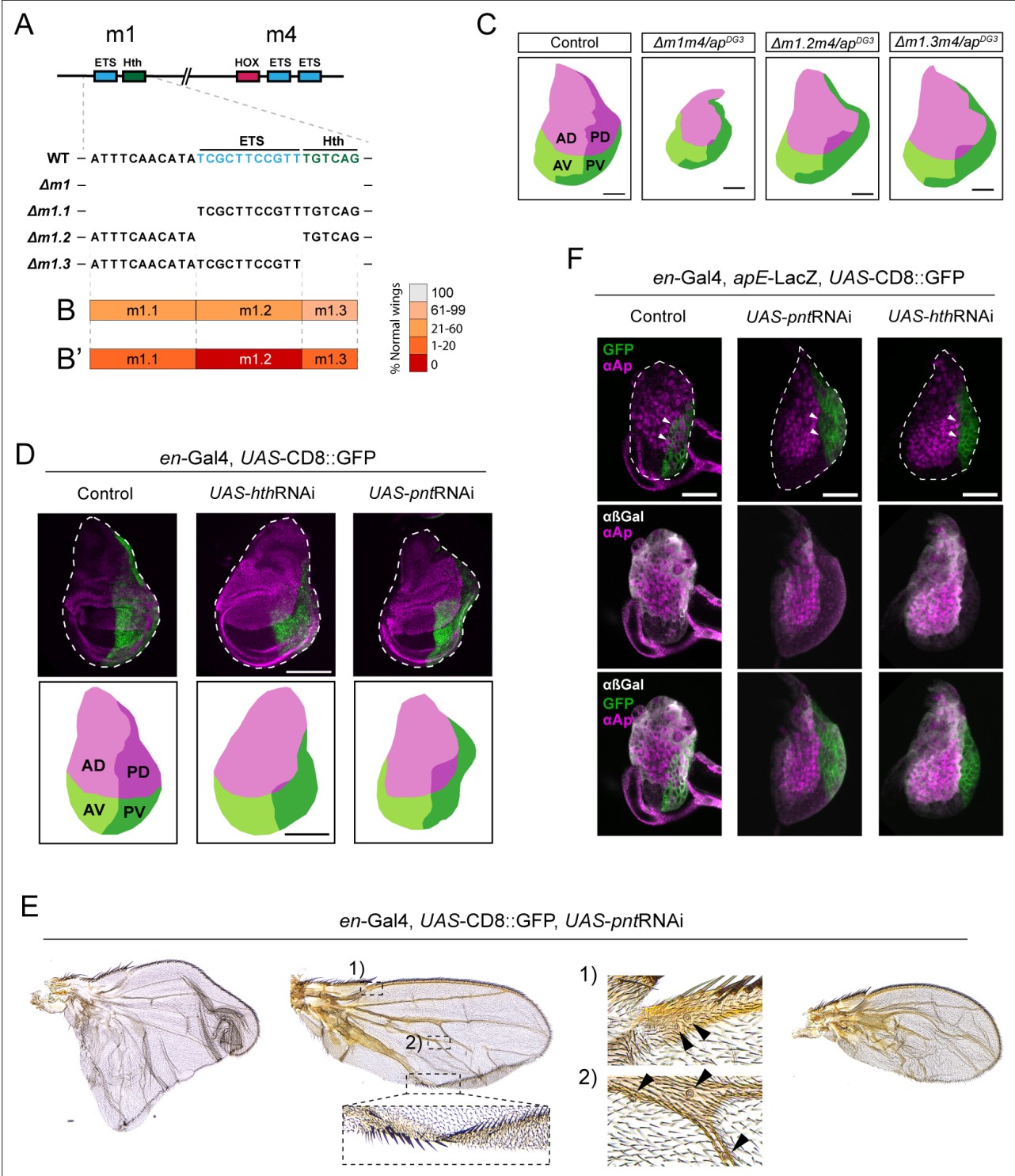

**Figure 4.** Pnt and Hth are required for apE activity via m1. (**A**) Schematic representation of m1 and m4 predicted binding sites and the generated deletions within m1. (**B**) Phenotypic penetrance of the different deletions within m1 in homozygotes, and hemizygotes (**B′**). (**C**) Projected area of the different quadrants of control WT/ap^DG3^, Δm1/apDG^DG3^, Δm1.2/apDG^DG3^, and Δm1.3/apDG^DG3^ wing discs, based on the immunostaining against Ap and Ptc in real wing discs. (**D**) Effect on the Ap expression domain by the expression of *hthRNAi* and *pntRNAi* in the P domain using *en*-Gal4. In both cases the PD domain is reduced. In green, the *UAS*-CD8:GFP reporter marks the P compartment. (**E**) Adult wing phenotypes upon *pntRNAi* expression via *en*-Gal4. Left wing: Example of a wing outgrowth resembling Δm1 phenotype. Middle wing: Example of a partial A mirror-image transformation. Inset 1: Campaniform sensillae of the A compartment. Inset 2: Ectopic campaniform sensillae formed in the P compartment upon *pnt*RNAi expression. Notice the presence of A bristles within the P compartment. Right wing: Example in which P compartment is reduced. (**F**) Effect of the expression of *hthRNAi*

*Figure 4 continued on next page*

*Figure 4 continued*

and *pntRNAi* in the P domain on anti-Ap localization and apE-LacZ reporter (visualized with anti-βGal) during L2 stage. In control discs, anti-Ap can be detected in the nucleus of P cells (marked with *UAS*-CD8:GFP) (arrowheads). anti-βGal (in white) is detected in the dorsal compartment in both A and P compartments. Upon *pntRNAi* or *hthRNAi*, no anti-Ap nor anti-βGal signal was detected in P cells (arrowheads). Scale bars: panels C and D, 100 μm; panel E, 25 μm.

To address the role of these TFs in apE activation, we tested the effect of the *pnt* and *hth* knock-down on the expression of an apE-LacZ reporter fusion. apE is mainly expressed during early larval stages. Thus, we choose mid L2 wing discs for this comparison. As before, *UAS*-CD8:GFP was used to label the manipulated cells. In these early discs, both Ap staining and apE-LacZ reporter were strongly reduced upon *pnt*RNAi expression (*Figure 4F*). In the case of *hth*RNAi, we observed a total abrogation of both (*Figure 4F*). Together, these results strongly indicate that both Hth and Pnt are required for early *ap* expression via m1.

## High-resolution m3 analysis suggests a GATA–HOX complex required for wing development

Our mutation analysis revealed that the most sensitive region of OR463 is m3 (*Figure 1*). Compared to controls, Ap expression was reduced to a small group of cells in the hinge upon *Δm3* deletion (*Figure 5A*). As expected from the adult phenotypes, these mutants lacked the wing pouch, with the inner ring of Wg expression reduced to a single point (*Figure 5A'*). Enhancer activity was then tested using apE-LacZ reporters. Despite apE-LacZ being expressed mainly during early larvae development (*Bieli et al., 2015b*), LacZ is highly stable within cells, which permits, upon long exposure to X-Gal, the rough evaluation of enhancer activity in L3 wing discs (*Figure 5B*). Deletion of m3 in this setting dramatically reduced X-Gal staining (*Figure 5B'*), indicating that the factors binding m3 are indispensable for apE enhancer activity.

Bioinformatic analysis of m3 revealed two highly conserved DNA regions, recognized by putative GATA and HOX TFs. To avoid possible artifacts derived from the sequence rearrangement that deletions produce, we generated a library of fly strains bearing precise nucleotide substitutions in pairs (G → T, A → C, T → C, and C → A), allowing the evaluation of the functional dependency of each base pair within the region in the endogenous setting (*Figure 5C*). Phenotypic scoring revealed that the sequences associated with both the putative GATA- and HOX-binding sites were required for normal wing development (*Figure 5D*). Mutations within the GATA-binding site resulted in phenotypes resembling those described for other OR463 mutants (*Figure 5E'*). Mutations affecting the HOX-binding site were comparatively more severe and more penetrant (*Figure 5E'''*). Four of the lines carrying mutations within the predicted HOX-binding site yielded no wings (*Figure 5E''*), indicating a strong requirement of this putative binding site for wing development. The same results were obtained when the binding sites were deleted individually (*Figure 5—figure supplement 1A, B*). Deletion of the predicted HOX-binding site resulted in the same Ap distribution as the complete deletion of m3 (*Figure 5—figure supplement 1C*).

The very high conservation of the two sites suggests that both factors may interact during enhancer activation. To genetically test this hypothesis, we designed a series of mutants in which we modified the distance between these binding sites. The normal spacing was either expanded to 6 bp or contracted to 2 bp (*Figure 5F*, *Figure 5—figure supplement 2A*). When spacing was increased, all wings presented defects and, in 28% of the cases, wings were totally missing (n=110) (*Figure 5—figure supplement 2B*). Reduction of the spacer sequence resulted in comparatively milder phenotypes, yielding 30% of normal wings and most of the other having a posterior compartment outgrowth (*n* = 155) (*Figure 5—figure supplement 2C*). These results suggest that the distance between these TFs is important for the correct functioning of the enhancer, likely implicating a previously unknown GATA–HOX complex involved in wing development.

m3 bears a strong sequence identity with m2. In fact, both regions are predicted to contain one GATA- and one HOX-binding sites. In contrast to the 5-bp separation between these sites in the m3 region, the m2-binding sites are separated by 11 bp. As an independent test for the requirement of correct spacing, we evaluated whether reducing the distance of m2-binding sites could rescue the lack of m3, whose deletion results in fully penetrant wing loss (*Figure 1G*, *Figure 5—figure supplement 2E*). Interestingly, reducing the spacing of m2 in *Δm3* mutants reduced to 69%

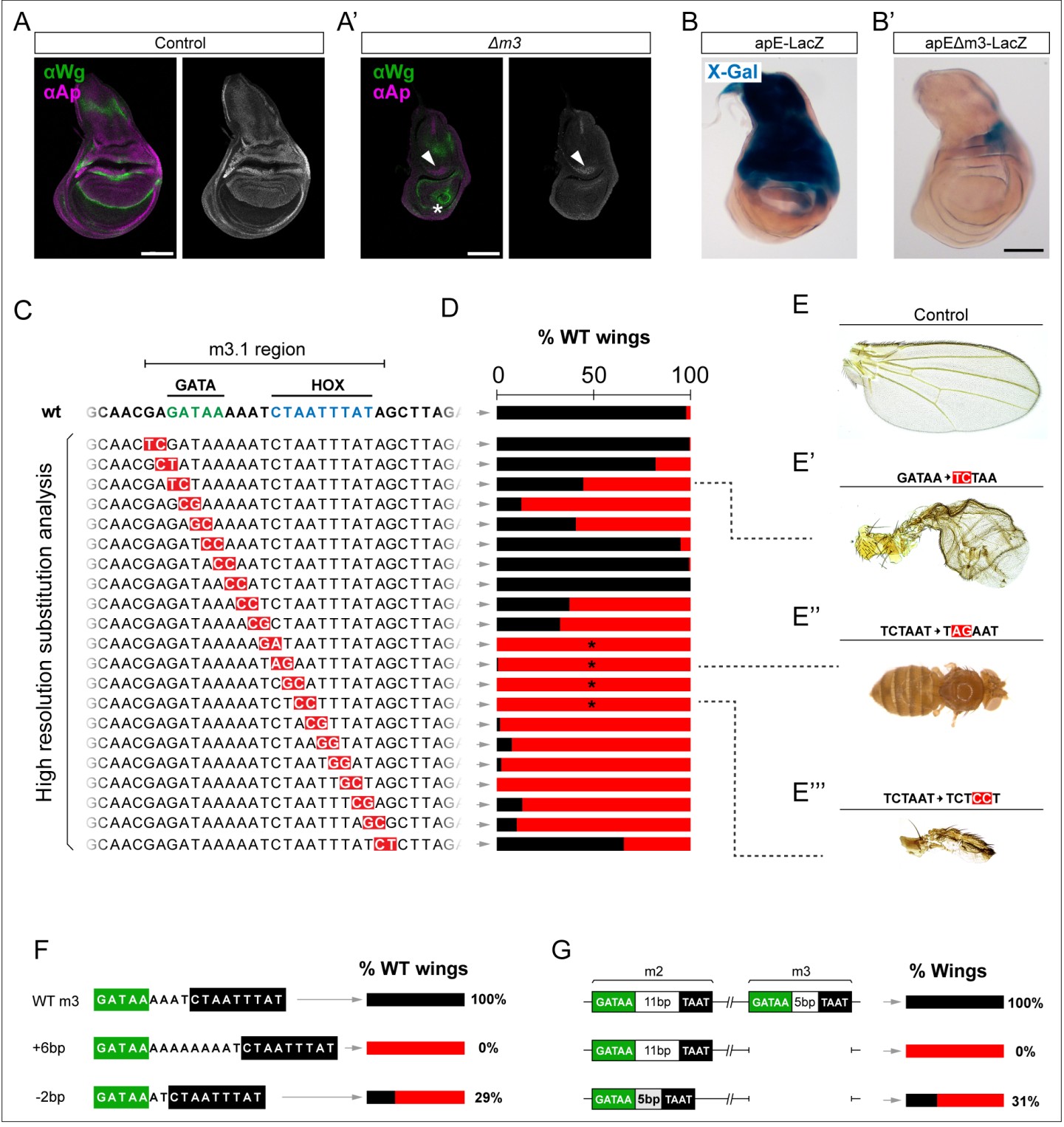

**Figure 5.** High-resolution genetic analysis suggests a GATA–HOX complex important for m3 activity. (**A**) Anti-Ap and anti-Wg immunostaining in control and Δm3 third instar wing discs. In contrast to control wing discs, Ap can only be detected in a small group of cells in the anterior hinge in Δm3 mutants (arrowhead). The territory within Wg inner ring is totally missing (asterisk). No Wg stripe is detected in the pouch. (**B**) X-Gal staining of control apE-LacZ and apEΔm3-LacZ in third instar wing disc. apEΔm3-LacZ only showing minimal X-Gal staining in the P hinge. (**C**) Summary of the base-pair substitutions generated within m3.1. Each row corresponds to a different allele containing the changes labeled in red. (**D**) Scoring of wild-type wings across the library of Δm3 mutants. In black, the percentage of WT wings. Between 80 and 250 wings were scored in each case. Asterisks denote mutants that gave rise to no wings with different penetrance. (**E–E'''**). Wing phenotypes of the control, and three of the mutants of the library. (**F**) Phenotypic penetrance scoring

*Figure 5 continued on next page*

*Figure 5 continued*

of control animals and individuals in which the linker between GATA- and HOX-binding sites was extended or contracted. (**G**) Percentage of flies with wings upon deletion of m3, or deletion of m3 and simultaneous contraction of m2 linker. Scale bars: 100 μm.

The online version of this article includes the following figure supplement(s) for figure 5:

**Figure supplement 1.** Deletion analysis within m3 region.

**Figure supplement 2.** Correct spacing between GATA- and HOX-binding sites is essential.

the loss of wings (*Figure 5G*). The most frequent phenotype among the remaining 31% were tube-like wings (*Figure 5—figure supplement 2F′*). In rare cases, more normal wings were observed (*Figure 5—figure supplement 2F″*). This small but significant rescue further supports the hypothesis that the interaction between a GATA and a HOX factor upon DNA binding is important for OR463 activity.

## Grn is fundamental for wing development

The *Drosophila* genome contains five genes encoding GATA TFs: *pannier (pnr)*, *grain (grn)*, *GATAd*, *GATAe*, and *serpent (srp)*. So far, only *pnr* has been involved in wing disc development (*García-García et al., 1999*); however, its expression pattern is constrained to the most proximal region (*Ramain et al., 1993*), with its loss of function being associated with defects in the notum. *grn* expression has been detected in the wing hinge and pouch (*Brown and Castelli-Gair Hombría, 2000*; *Everetts et al., 2021*). However, mutant clonal analysis yielded no phenotypes in this tissue (*Brown and Castelli-Gair Hombría, 2000*).

We knocked-down *grn* via RNAi expression in the P compartment using *en*-Gal4. *UAS*-GFP was used to mark the manipulated cells. To increase knock-down efficiency, we included *UAS-dicer* in all crosses. Among the four *grn*RNAi transgenes available from Bloomington, a test experiment indicated that two of them were suitable for our purposes (the other two had no phenotypic consequences upon expression). *grn*RNAi BL34014, when combined with *UAS-dicer; en-Gal4, UAS-GFP*, produced flies with abnormal notum and/or wings at low frequency (11%). Among the phenotypes, we observed loss of the wing or notum and wings with anterior image duplications, as well as others displaying reduced P compartment size (*Figure 6A*). The other *grn*RNAi stock (BL27658), when expressed with the same driver line, led to developmental arrest in white pupae. This line was used to study the effect of *grn* depletion in the wing imaginal discs. This experiment revealed a dramatic defect in P growth compared to control discs (*Figure 6B*). In all cases, P cells did not show detectable Ap expression. As a result, the P compartments did not remain associated with the DA cells, positioning them randomly within the rest of the disc. This 'floating' nature of the P compartment is likely to explain the diversity of adult phenotypes observed with the *grn*RNAi BL34014 transgene. In some cases, the P compartment was located close to the DV boundary, promoting some degree of pouch growth but leading to partial or total notum loss (*Figure 6B′*). In other cases, P cells ended up in the notum region, leading to somewhat proper patterning of this region but resulting in dramatic defects of the hinge and pouch (*Figure 6B″*).

To independently validate these results, we generated tissue-specific CRISPR knockouts of *grn*. Ubiquitously expressed *grn* gRNAs (*U6-grn.gRNAs*) were combined with *UAS*-Cas9 under the control of en-Gal4. P cells were marked using *UAS*-CD8:GFP. In these cases, the P compartment was totally depleted in mid L3 wing discs (*Figure 6C*), supporting the results obtained with RNAi. In order to study *ap* expression in these discs, we decided to analyze discs at an earlier time point (mid L2). At this time, P cells had not been lost and GFP-expressing cells still occupied a considerable area of the wing. Ap protein was not detected in P cells (*Figure 6D*). Expression of the apE-LacZ reporter was also abrogated (*Figure 6D*), suggesting that *grn* is required for *ap* transcriptional activation via apE.

*grn* expression during wing disc development was then analyzed using a *grn*^GFP transgene that contains all the cis-regulatory information (*Kudron et al., 2018*). In agreement with previous reports (*Brown and Castelli-Gair Hombría, 2000*), Grn:GFP was detected throughout the wing disc already in very early larval stages, but mainly in the hinge, where it was enriched in the A compartment. In addition, we observed that the area with the highest Grn:GFP signal was located around the contact between the tracheal system and the disc proper (*Figure 6—figure supplement 1*).

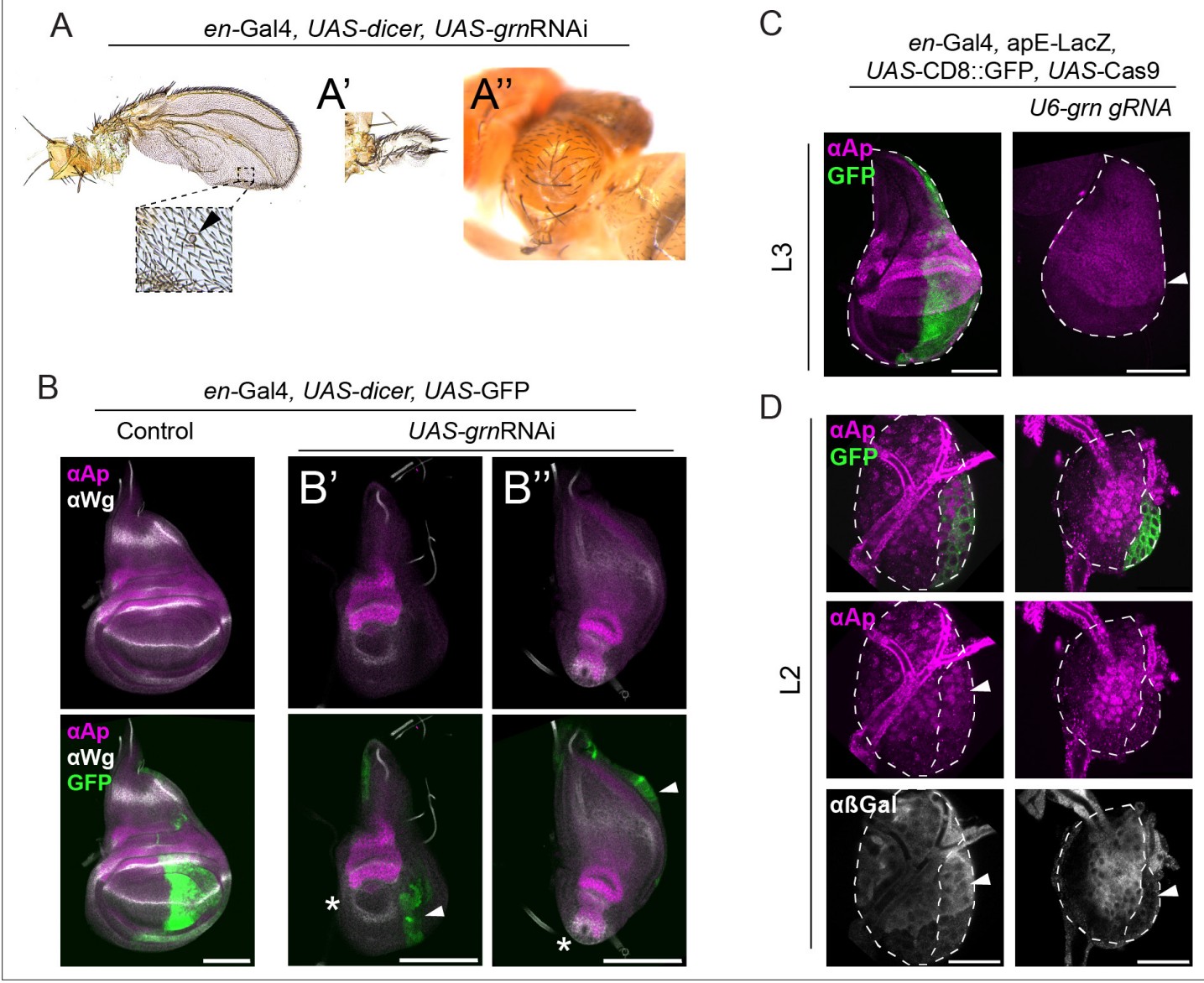

**Figure 6.** The GATA TF Grain is required for wing development. (**A**) Phenotypes produced upon *UAS-grnRNAi* expression driven by en-Gal4 in the presence of *UAS*-dicer. Reduction of the P size and partial P to A transformation (evidenced by the campaniform sensilla in the P territory, arrowhead). (**A'**) Mirror-image duplication of an anterior proximal rudimentary wing structure. (**A''**) Thorax defects observed in some of the adults. (**B**) Anti-Ap and anti-Wg immunostaining of control wing discs and P knockdown of *grn* (*grn*RNAi driven by en-Gal4). In both cases, *UAS*-GFP was used to mark P cells and *UAS*-dicer was included to increase knockdown efficiency. (**B'**) Example of wing disc in which the P compartment (arrowhead) was located close to the DV boundary. In these cases, the pouch was specified and grew to some extent, as revealed by the space within the inner ring of Wg (asterisk). (**B''**) Example of wing disc in which the remnant of the P compartment (arrowhead) is located close to the notum. Here, the notum primordium grew to a considerable size and presented the characteristic Wg band, indicating, to some extent, correct patterning. The pouch (asterisk) is completely absent. (**C**) Anti-Ap immunostaining upon tissue-specific knock-out of *grn* in the P cells using *en*-Gal4, UAS-Cas9 in the presence of *U6-grn*.gRNAs. *UAS*-CD8:GFP labeled posterior cells. Notice the total loss of P compartment in mid L3 wing discs (illustrated by the complete absence of GFP signal). (**D**) L2 wing disc of the same genotype as in C. Immunostaining of apE-LacZ using anti-β-Gal, as well as anti-Ap reveals total lack of signal in the P cells (arrow heads). Scale bars: panels B and C, 100 μm; panel D, 25 μm.

The online version of this article includes the following figure supplement(s) for figure 6:

**Figure supplement 1.** Pattern of Grn:GFP localization during wing development.

## Antennapedia is indispensable for wing development and *ap* expression

Our results demonstrated that there is a single putative Homeodomain-DNA interaction in OR463 that is indispensable for wing development (*Figure 5E"*). However, the homeodomain-containing protein binding to this site remains to be characterized. The fly genome contains 103 genes that encode proteins with homeodomains (*Bürglin and Affolter, 2016*). Among those, Antennapedia (Antp) emerged as a plausible candidate, as it is responsible for the embryonic identity of the segment T2, from which the wing discs derive. Supporting an instructive role in wing formation, *Antp* overexpression specifies ectopic wings in the head under certain genetic conditions (*Kurata et al., 2000*; *Prince et al., 2008*). Recent reports demonstrated that *Antp* is required during larval stages for wing disc morphogenesis and wing margin development (*Paul et al., 2021*). However, the wings presented in this study showed mild phenotypes compared to the total loss of wings seen in *Δm3* mutants. We thus hypothesized that Antp may contribute to the early activation of the enhancer. Such an early role of Antp in wing disc development could have been easily missed in previous studies, which only depleted *Antp* during larval stages. Consistent with a potential early role for Antp in wing development, early *Antp* MARCM clones generated during the first 24 hr of development were never retrieved in larval stages in the disc proper, and only very rarely in the peripodial membrane (*Figure 7—figure supplement 1A*). Interestingly, clones generated at 48 hr in development did not affect anti-Ap levels (*Figure 7—figure supplement 1B*), indicating that *Antp* is dispensable for *ap* expression in mid-late larval stages.

To deplete *Antp* during embryonic development, tissue-specific CRISPR/Cas9 was employed. First, we designed flies expressing ubiquitously expressed gRNAs targeting both the first protein-coding exon of *Antp* and its third protein-coding exon (which encodes the homeodomain) (*Figure 7A*). Second, to target the wing primordium as early as the disc is specified, we employed the driver *sna*1.7-Gal4, specifically expressed in the embryonic primordia of the wing and haltere from stage 12 onwards (*Figure 7A*; *Requena et al., 2017*). Control experiments showed robust Antp immunoreactivity in the wing disc, with its highest levels in the hinge and A compartment (*Figure 7B*), in agreement with previous reports (*Levine et al., 1983*; *Paul et al., 2021*). In contrast, when Cas9 was expressed via *sna*1.7-Gal4 in the presence of *Antp* gRNAs, Antp immunoreactivity was severely affected, yet was only totally depleted in some discs (*Figure 7C*). Interestingly, some of these discs presented dramatic changes in the Ap pattern, presenting only isolated Ap-expressing clones (*Figure 7C'*). anti-Ap intensity in these clones was comparable to controls, suggesting problems in *ap* early pattern establishment rather than later maintenance of *ap* expression. Finally, in those discs without Antp immunoreactivity, no Ap was detected (*Figure 7C"*). These discs presented severe malformations and did not exhibit the normal notum-hinge-pouch structure of wild-type wing discs. Many of these individuals died during development, with survivors presenting problems in notum closure (data not shown). Among those flies that hatched, most flies presented correctly patterned wings (*Figure 7D*), although they remained inflated in many cases. Some of the most severely affected flies lacked most wing disc derivatives, including the wings and parts of the notum (*Figure 7E*). Halteres were always normal (*Figure 7E*), suggesting that the malformations arise due to the specific loss of *Antp* and not due to Cas9 expression. The poor viability of the *sna*1.7-Gal4, *U6-Antp*.gRNAs flies suggests that the individuals that hatched correspond to those wing discs that were least affected by gRNA expression (*Figure 7C*).

These results support a model in which Antp contributes to early wing disc development, potentially acting, at least in part, through the early activation of ap.

## Discussion

### Boundary position in the wing imaginal disc

Boundary positioning is essential for proper development (*Dahmann et al., 2011*; *Irvine and Rauskolb, 2001*). Mispositioning of developmental boundaries leads to severe developmental defects. In the wing imaginal disc, the intersection between two developmental boundaries is responsible for the specification of the wing pouch and the establishment of the coordinates which regulate tissue patterning. In this study, we analyzed the role of the apE enhancer in the correct establishment of the DV boundary. In previous studies, it had been shown that the EGFR pathway is a key input required

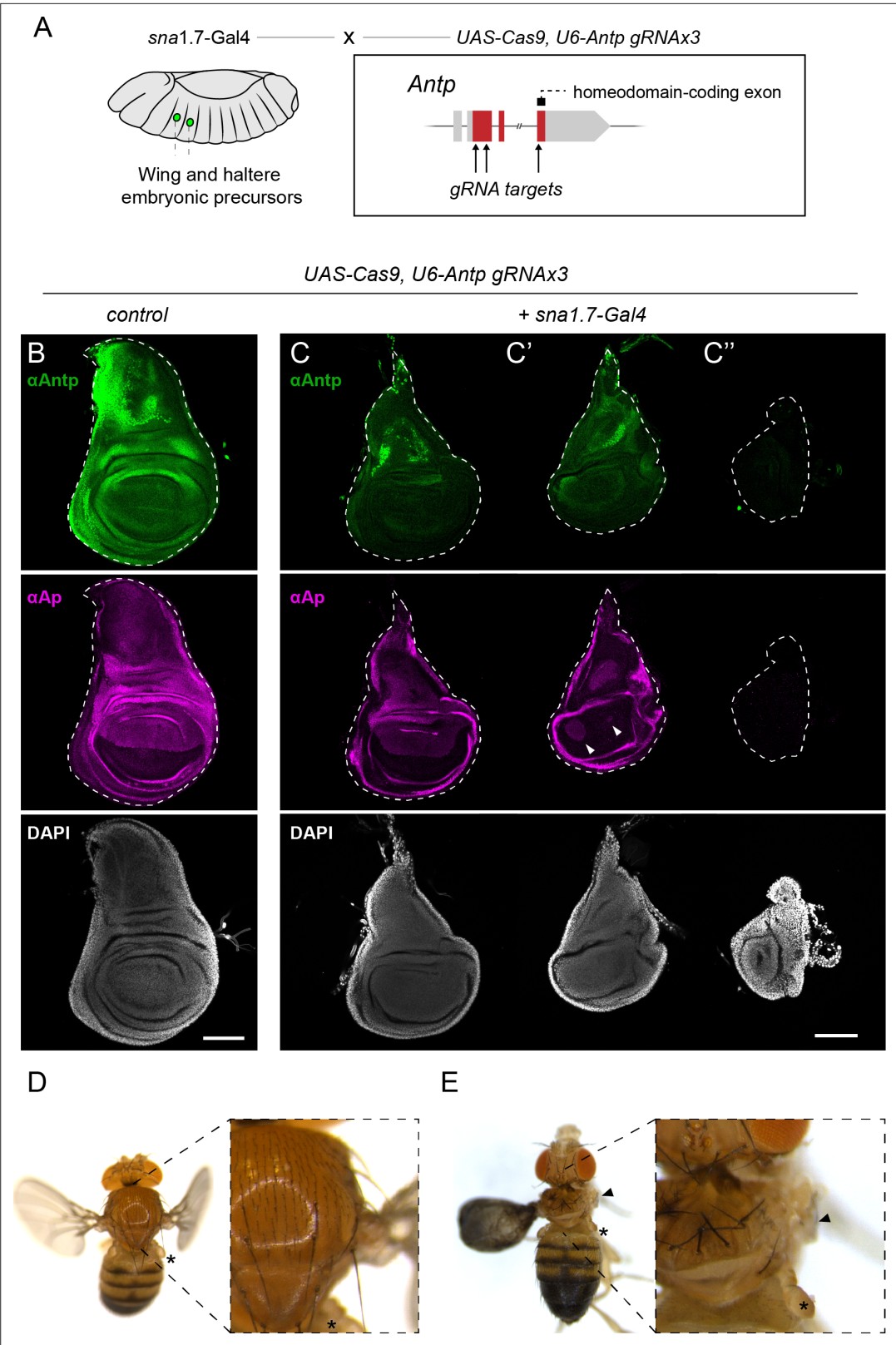

**Figure 7.** The HOX gene *Antp* is fundamental for early wing development and *ap* expression. (**A**) Scheme of the experimental setup to delete *Antp* during early stages of wing development. *sna*1.7-Gal4 driver is used to express Cas9 in the embryonic precursors of wing and haltere. Cas9 is targeted to the *Antp* locus by three gRNAs (labeled with an arrow). Position of the homeodomain within *Antp* sequence is also indicated. (**B**) Anti-Antp and

*Figure 7 continued on next page*

*Figure 7 continued*

anti-Ap immunostaining in control third instar wing discs. Notice the distribution of Antp throughout the disc, with its highest levels in the A compartment and hinge, and the faint band along the DV boundary. (**C**) Example of wing disc derived from *sna*1.7-Gal4 driven Cas9 in the presence of *U6-Antp*.gRNAs. In this case Ap presented a rather normal distribution in the tissue. Antp could still be detected in the wing discs, amid a reduction in its levels. (**C'**) Wing disc of the same genotype as C presenting severe problems in the Ap expression pattern. Ap was detected in the pouch only in two groups of cells (marked with an arrowhead). (**C''**) Example of a wing disc of the same genotype as C and C'' in which no immunoreactivity against Antp is detected. Note the complete lack of the anti-Ap signal. In gray, DAPI. These discs lack all recognizable structures (no notum, and no pouch). (**D**) Adult phenotype arising from the same genotype as B (control). The haltere is marked by an asterisk. (**E**) Example of a severe case in which *Antp* was knocked out from the wing primordia (same genotype as in C, **C', and C''**). In this case, only a small portion of the right notum is still present, with the total loss of the right wing (arrowhead). Left wing presents severe morphological defects and forms a balloon-like structure. Further analysis of this wing revealed the presence of campaniform sensillae and A bristles in the P compartment (data not shown). Notice that the halteres were unaffected (asterisk). Scale bars: 100 µm.

The online version of this article includes the following figure supplement(s) for figure 7:

**Figure supplement 1.** *Antp* is required for clone survival in early stages and does not affect *ap* expression at later stages.

for early *ap* transcription (*Bieli et al., 2015b*; *Wang et al., 2000*; *Zecca and Struhl, 2002*). However, the expression pattern of the EGFR ligand *vn* is highly dynamic during the onset of *ap* expression: in L1 and L2 larvae, *vn* is expressed in a proximodistal stripe, both in future ventral and dorsal cells, which only later is confined to the proximal end of the wing disc (*Paul et al., 2013*). In contrast, *ap* is first expressed in the center of the wing disc, acquiring its characteristic pattern during the second instar (*Nienhaus et al., 2012*; *Paul et al., 2013*). Together, these patterns suggest a more complex regulatory landscape. In this study, we provide evidence that both Hth and Grn are required for apE activity (*Figures 4D and 7*). Interestingly, these two factors are expressed early in the presumptive hinge region (*Brown and Castelli-Gair Hombría, 2000*; *Wu and Cohen, 2002*) (see *Figure 8* for a working model on *ap* regulatory inputs). In particular, Grn is enriched in the areas in which the transverse connective trachea is in contact with the wing (*Guha et al., 2009*). This extra layer of apE regulation might explain why the early expression of *ap* is initially confined to the central part of the wing, and why the first cells expressing *ap* are often located close to the trachea (*Nienhaus et al., 2012*). Remarkably, both Hth and Grn are also expressed in the ventral hinge, which implies that either these factors are not sufficient to activate *ap* expression or that their activity is actively repressed in this territory. Along these lines, the TF Optomotor-blind (Omb) has recently been proposed to repress *ap* expression in the distal wing (*Chen et al., 2023*).

Loss of EGFR signaling has also been associated with mirror-image duplications (*Baonza et al., 2000*). However, the mechanism proposed was that EGFR maintains *en* expression in P cells. Interestingly,

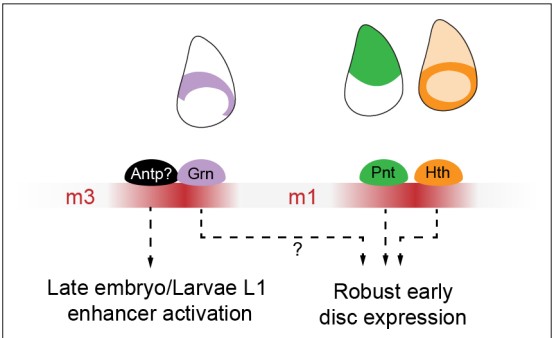

**Figure 8.** Working model for OR463 regulatory inputs. During late embryonic or early larval stages, the HOX input in m3, potentially mediated by Antp, would be responsible for OR463 activation. During this early phase the enhancer is not yet functional and the HOX could be priming the enhancer, permitting the later action of the other factors. Grain would participate in this process but its requirement is less critical than that of the HOX. Pnt and Hth would then, during L2 larval stage, activate apE first in the proximal area of the wing disc. Grn could also play a role in this early activity, confining the activity of *ap* to the dorsal hinge.

many of the duplications reported in this study mimicked those presented here (*Figure 1*). Given that the EGFR pathway activates apE (*Bieli et al., 2015b*) and that the deletion of Pnt-binding sites within apE also result in mirror-image duplications (*Figure 4B*), we propose that the previously reported phenotypes arise from apE misregulation rather than from loss of *en* maintenance. To confirm whether this is the case, the lineage of *en*-expressing cells should be analyzed upon EGFR loss.

Furthermore, downregulation of JAK–STAT signaling has been shown to produce mirror-image duplications (*Recasens-Alvarez et al., 2017*). The OR463 phenotypes described in *Figure 1* also mimic those of JAK–STAT defects. *ap* is fundamental for the development of the hinge, where Upd-1, the ligand of the JAK–STAT pathway, is expressed (*Johnstone et al., 2013*). Thus, *ap* expression may indirectly disrupt JAK–STAT signaling by impairing ligand production. Alternatively, JAK–STAT mirror-image duplications could arise from direct regulation of apE. Interestingly, some STAT92E-binding sites are predicted in N1 and N2 (data not shown). It is also possible that direct regulation of apE via JAK–STAT signaling contributes to these duplications.

Although apE is active throughout the dorsal compartment, its disruption leads to a preferential loss of ap expression in posterior cells. The asymmetric effect of apE perturbation on the anterior and posterior compartments suggests that apE transcriptional control is not equivalent across the AP axis. Compartment-dependent differences in enhancer regulation have also been documented in other developmental contexts; for example, the Distal-less DMX-R element is interpreted through distinct cofactor combinations (Sloppy paired anteriorly and Engrailed posteriorly) (*Gebelein et al., 2004*), and specific mutations within DMX-R preferentially disrupt enhancer function in anterior versus posterior cells. It is possible that apE is more sensitive to misregulation due to differential transcriptional regulation across compartments. Nevertheless, we cannot exclude the possibility that the posterior bias we observe arises not from enhancer logic per se, but from intrinsic differences in tissue architecture or the dynamics of boundary positioning during wing disc development.

## A GATA–HOX complex involved in wing development

The genetic control of wing specification is well studied and many of the factors involved have been characterized. With the exception of the JAK-STAT pathway (*Recasens-Alvarez et al., 2017*), most of the factors required for wing specification were identified more than two decades ago. Here, our bottom-up approach permitted us to describe the requirement of two additional factors for the development of the wing imaginal disc. Our data suggests that the GATA TF Grn is fundamental for wing development. *grn* was originally described as a gene involved in epithelial morphogenesis in the embryo and the legs. Here, we report that loss of *grn* also causes extreme phenotypes in the wing disc (*Figure 6*). Interestingly, we found that *grn* is not only important for activation of *ap*, but also for the growth and survival of the whole tissue. P compartments lacking *grn* were of very small size or totally lacking (*Figure 6*). However, their elimination occurred over a long period, during which the mutant cells were still able to instruct patterning and growth to their surroundings. Previously, mutant clones of *grn* covering the wing blade were reported not to produce any developmental defects (*Brown and Castelli-Gair Hombría, 2000*). The apparent difference between these results is likely to emerge from the different approaches used to eliminate *grn* function. While the previous study performed a clonal analysis using the minute technique, here we knocked down *grn* using RNAi or knocked it out using a tissue-specific CRISPR. These two approaches resulted in a homogeneous loss of *grn* function in all posterior cells already in early embryonic stages, which is not the case for Minute clones. It is likely that this temporal difference is the main reason for these discrepancies, as we have also observed that *grn* clones generated late in development do not affect *ap* expression (data not shown), suggesting that Grn is only required during very early stages of wing development.

Most point mutations or short deletions in enhancer regions have little effect on gene expression. This robustness emerges, in part, from the collective action of TFs, whose binding sites are often clustered in groups. In addition, the deletion of enhancers which were thought to have central roles in development has often been found to have minor phenotypes due to existence of redundant sequences located in the vicinity (*López-Rivera et al., 2020*). Even in *Drosophila*, examples of mutant phenotypes derived from point mutations in enhancers are rare (*Shimell et al., 1994*). Recent large-scale enhancer mutagenesis studies have shown that the mutational consequences within enhancers can vary widely. In some cases, many nucleotide positions appear tolerant to single-base changes and only a small subset of mutations produce clear functional effects (*Kvon et al., 2020*). In other

enhancers, regulatory information is distributed more densely, and mutations at multiple positions can alter output (**Fuqua et al., 2020**). Together, these studies illustrate that enhancer sensitivity is not uniform but depends on enhancer-specific features such as motif organization, cooperativity, and redundancy. Within this broader landscape, the *apE* enhancer appears to be unusually sensitive to mutation.

In this study, we have identified an area of the *Drosophila* genome with exceptional sensitivity to mutation. Four of our two-base-pair substitutions within the HOX-binding site of m3 resulted in total loss of wings. In this study we propose that Antp is a strong candidate HOX TF acting at m3. In support of this hypothesis, we demonstrated that early loss of *Antp* in the embryonic wing primordium results in defects in *ap* expression (**Figure 7C'**), in severe cases leading to a total loss of Ap (**Figure 7C"**). Until recently, the second thoracic segment was assumed to develop independently from any HOX input. However, several groups have now reported that this segment, from which the wing is derived, requires the input of the HOX gene *Antennapedia* (*Antp*) for its development (**Fang et al., 2021**; **Paul et al., 2021**). These studies uncover a role of Antp during larval wing morphogenesis and wing margin formation. Given the severity of the phenotypes we observed (**Figure 7C", E**), our experiments are consistent with a previously underappreciated early role of this gene, most likely during wing disc specification. While we demonstrate the requirement of Antp for *ap* expression, whether this effect is direct and mediated through the m3-binding site remains to be established biochemically. HOX TFs have been proposed to function as pioneer TFs in some contexts, changing the enhancer's chromatin accessibility and enabling the binding of other TFs to the enhancer (**Bulajić et al., 2020**; **Loker et al., 2021**). Such pioneer Antp activity could explain the severity of the mutations in apE. However, biochemical characterization of the locus in early wing imaginal discs precursor cells is required to test this hypothesis.

The current working model for the functioning of OR463 includes two steps. First, Grn and a HOX TF, possibly Antp, would act on the enhancer, permitting its later activity. Then, during L1 and L2 stages, Pnt and Hth, probably together with other TFs, would confer the spatial clues that result in the early Ap activity. The mechanism of action of each factor at the enhancer remains to be investigated.

It is important to acknowledge that all expression analyses were conducted in third-instar discs, a stage that follows the initial establishment of *ap* expression. Earlier effects are therefore inferred rather than directly observed, as imaging and staging of early discs present significant technical challenges due to their small size and fragility. Direct observation of the early wing discs across mutant conditions would help clarify the role of the discovered factors during early ap expression.

## New method to manipulate enhancers

In the past decades, the functional analysis of enhancers depended on the generation and analysis of lesions in the respective DNA fragment, often studied with reporter transgenes, in which the mutated enhancer was fused to a minimal promoter driving the expression of a reporter gene (LacZ, GFP, etc.). Such analyses provide an idea of the spatiotemporal changes in enhancer function when the enhancer is mutated. Minimal reporter fusions have been fundamental to characterize enhancers; however, follow up studies of such mutation analyses using endogenous gene editing often demonstrates minor effect for many mutations when analyzed in their endogenous genetic environment. Moreover, this approach does not permit to manipulate and characterize the requirement of a particular binding site in time and space. In recent years, the advance of techniques based on catalytically dead Cas9 (dCas9) has opened the door for the precise somatic disruption of enhancers. Enhancer repression has been mainly accomplished by dCas9-mediated recruitment of epigenetic repressors (**Wang et al., 2016**). However, the precision of this approach is hard to measure and depends on the repressive element to which dCas9 is fused. dCas9 binding to specific sites has also been shown to be sufficient to disrupt TF-DNA binding in prokaryotes (**Qi et al., 2013**) and in cell culture (**Shariati et al., 2019**), opening the door to a more precise manipulation of selected DNA regions or sites. Here, we adapted the latter approach for its use in vivo in *Drosophila*. We demonstrate that the sole expression of dCas9 together with gRNAs binding OR463 is sufficient to mimic the phenotypes caused by endogenous enhancer mutants. Moreover, the localized expression of dCas9 permitted us to show that the mirror-image duplications arise due to the early defects in the activity of the enhancer in the P compartment and to delimit the temporal window of enhancer function.

This approach has broad applicability and may be useful to investigate the importance of regulatory sequences across the *Drosophila* genome. The ease by which gRNA expressing stocks are generated (*Port and Bullock, 2016*) and the possibility to encode many gRNAs in a single transcript will further simplify the functional characterization of enhancer regions.

## Materials and methods

### Fly strains and husbandry

The following fly strains were used in this study: *vasa*-Cas9 (Bloomington number: BL51323), *ap^DG3^* (*Bieli et al., 2015a*), *ap^DG12^* (*Bieli et al., 2015b*), *en*-Gal4 (gift from Manuel Calleja), *ptc*-Gal4 (BL2017), *sna*1.7-Gal4 (*Requena et al., 2017*), G-TRACE (BL28281), *UAS*-CD8:GFP (BL32186), *upd1*-Gal4 (gift from Prof. Denise Montell), *UAS*-EGFP (BL6874), UAS-*pnt*RNAi (BL35038), UAS-*hth*RNAi (BL34637), UAS-*grn*RNAi (BL27658 and BL34014), *UAS-dicer; en*-Gal4, UAS-2xEGFP (BL25752), apE-LacZ (*Bieli et al., 2015b*), U6-*grn*.gRNA (BL85866), *UAS*.Cas9 (BL58986), *grn*:GFP (BL585483), *tub*-Gal80^ts^ (BL7016). The stocks generated in this study are listed in *Supplementary file 2*. Flies were kept at 25°C unless otherwise specified and were fed using home-made polenta supplemented with yeast.

### Immunostainings and image acquisition

For immunostaining, larvae of the desired stage were dissected in ice cold PBS (pH 7.2, Gibco) and immediately transferred into paraformaldehyde (PFA) solution (4% PFA in PBS) for 30 min at room temperature. The dissected tissues were washed 2 × 15 min with PBST (0.3% Triton X-100 in PBS) to permeabilize the tissue, followed by a 1-hr incubation in blocking solution (5% Normal Goat Serum in PBST). Subsequently, samples were incubated in primary antibody diluted in blocking solution at 4°C for 16 hr. The next day, the samples were washed 3 × 15 min with PBST and incubated for 2 hr in the secondary antibody diluted in blocking solution. Finally, samples were washed 3 × 15 min with PBST, followed by 2 × 15 min washes with PBS. All the steps were performed at room temperature using gentle rotation unless specified. Samples were mounted using Vectashield mounting medium (with or without DAPI) on a glass slide, covered with a coverslip (No. 1.5) and sealed using nail polish. Images were then acquired using a Zeiss LSM880 point confocal microscope and analyzed using ImageJ.

### Antibodies

The following primary antibodies were used in this study: anti-Ap (1/750) (*Bieli et al., 2015b*), anti-En (1/30, clone 4D9 DSHB), anti-Ptc (1/150, Apa1.3 clone, gift from Prof. Isabel Guerrero), anti-Hth (1/500, gift from Natalia Azpiazu), anti-β-Galactosidase (1/1000, Abcam AB9361), anti-Wg (1/120) (Clone 4D4, DSHB), anti-Antp (1/15, clone 8C11, DSHB), and anti-Vg (1/100, gift from Prof. Kristen Guss). The secondary antibodies employed were: goat anti-rabbit IgG (H+L) Alexa Fluor 680 (A-21109; Thermo Fisher), F(ab')₂ goat anti-rabbit IgG (H+L) Alexa Fluor 568 (A-21069; Thermo Fisher), goat anti-mouse IgG (H+L) Alexa Fluor 568 (A-11004; Thermo Fisher), Alexa Fluor 680 AffiniPure goat anti-mouse IgG, and Fcγ fragment specific (115-625-071; Jackson ImmunoResearch).

### Generation of OR463 landing site

Easy genetic screening of successful CRISPR/Cas9-mediated HDR-genome editing events is possible if an appropriate dominant selectable marker is available. Toward that end, we have adapted a feature of the dominant *apterous* allele *ap^Xasta^* (*Bieli et al., 2015a*). This allele is caused by a reciprocal translocation between the chromosome arms 2R and 3R that brings together the *ap* regulatory region and the *dad* (*daughters against dpp*) locus. *dad* is expressed in the wing disc in response to *dpp* signaling (*Weiss et al., 2010*). Due to the translocation, the wing-specific *dad* enhancers DadInt52 and Dad4 are located close to the *ap* promoter. This leads to ectopic *ap* expression in the ventral wing disc compartment that elicits the typical *Xasta* wing phenotype. Heterozygous flies show a dominant phenotype where the distal part of the wing is lost and a mitten shaped wing is formed. We have shown that when engineered into the *ap* locus, *dad* enhancers DadInt52 and Dad4 are sufficient to induce the dominant Xa-phenotype (*Bieli et al., 2015a*). Based on these observations, it was decided to clone a sub-fragment of Dad4, Dad13, into the donor plasmid used for CRISPR-mediated homologous recombination (HDR). Dad13 is only 517 bp long and has been shown to contain all necessary sequences for *dpp*-dependent activation (*Weiss et al., 2010*). Consequently, it was expected that

introduction of the Dad13 fragment into the *apterous* locus would induce a dominant wing phenotype. Thus, flies displaying this phenotype could be expected to contain successful homologous recombination events.

To generate the donor plasmid used as a template for HDR repair, a fragment containing the homology arms (each 536 bp long), an attP site (55 bp), two FRTs (34 bp each) flanking restriction sites AvrII and EcoRI and gRNA targets for linearization, was synthesized by Genewiz and cloned into pUC57-Kan. This plasmid is named pDB350. Subsequently, the 517-bp Dad13 enhancer was amplified from plasmid pDB45 using the primers 5′-CCCC<u>CCTAGG</u>CCTCAACCTTAAATTGTTG-3′ and 5′-CCCC<u>GAATTC</u>CGAACGGGAGAGCGCCGC-3′ and cloned by restriction ligation into the aforementioned plasmid using the sites AvrII and EcoRI (underlined). The final donor plasmid is called pMS358. In addition, two plasmids encoding gRNAs targeting OR463 were generated (pU6-m1-gRNA and pU6-m3-gRNA). The following primers were annealed and cloned into pU6-BbsI-chiRNA (*Gratz et al., 2013*) that was digested with BbsI: 5′-CTTCGCAGGCTCTGCCGAAGGTTCA-3′ together with 5′-AAACTGAA CCTTCGGCAGAGCCTGC-3′ and 5′-CTTCGTCAGAACGGTCAGCTACCTG-3′ together with 5′-AAAC CAGGTAGCTGACCGTTCTGAC-3′. gRNA targets were identified using CHOPCHOP (*Labun et al., 2019*).

After injection of the three plasmids into *y w vas-Cas9;+* embryos, surviving adults were individually crossed with *y w* flies. From these, 67 fertile F1 crosses could be screened for flies with wing defects. Successful HDR events could be isolated from four of the 67 G0 crosses (6%): c25, c77, c94, and c102. Alignment of the sequencing data with the in silico designed genomic sequence showed that candidates from all four crosses contained the expected changes down to the last base pair: the attP-FRT-Dad13 enhancer-FRT fragment is inserted in place of the 463-bp OR463 enhancer fragment. The sequence immediately flanking the inserted cassette (indicated by -[]-) is: CCCCTCGAATCGATTC AAAT-[]-TATAATTTGACCGCAATTTT. On the distal side, the break coincides with the *ap^{Xa}* break point.

As a result of the insertion of the Dad13 enhancer, it was expected that successful HDR-events would show an easy to spot phenotype similar to that in *ap^{Xasta}/+* flies. This was the case for some candidates isolated from cross c94. Although the phenotype is somewhat less pronounced, the tip of all wings always shows an obvious indentation (*Figure 1—figure supplement 3A′*). The dominant phenotype is much more subtle and less penetrant in other c94/+ candidate wings (*Figure 1—figure supplement 3A′*). In these and in candidates from the other three crosses, only a tiny spike of wing tissue emerges near the tip from the ventral side of the wing. We have previously observed a very similar phenotype in dpp-Gal4>UAS-dad flies (*Bieli et al., 2015a*). We have no explanation why insertion of the Dad13 enhancer gives rise to these different wing phenotypes.

From several of the candidates, the Dad13 enhancer was removed by Flp treatment. In all cases, homozygous *ΔDad13* derivatives were viable and fertile. Importantly, they also had no wings and halters, indicating that the designed deletion is a perfect starting point for our in-situ analysis of the apE wing enhancer. HDR-event c94.19 was chosen as landing site for mutated OR463 fragments and a *y w vas-integrase; c94.19^{ΔDad13}/CyO* stock was established. In this study, this landing site is referred to as *ap^{R2}*.

## Generation of OR463 rescue fragments

Most of the small deletions were engineered according to the method described by *Perez-Pinera et al., 2006*. A pBlueScript plasmid containing OR463 (pDB51) served as a template for the creation of these deletions. All primers used for this purpose are listed in *Supplementary file 3*. OR463 fragments containing more than one deletion (*Δm1m4* and *Δm1m2m4*) were generated step by step. At the end, all mutated OR463 fragments were amplified using the primers: 5′-TC<u>AAGCTT</u>TCCAGCTCTGAATCTCACTCCC-3′ and 5′-GT<u>ACTAGT</u>GGAGGCCAGCTCTAAAATCG -3′. Subsequently, amplified fragments were cloned into the re-entry plasmid pDB345 using the restriction sites HindIII and SpeI (underlined). All plasmids destined for transgenesis were verified by Sanger sequencing.

Several OR463 derivatives including the m3.1 substitution library were synthesized and cloned by Genewiz.

One of the deletions was obtained as a NHEJ-byproduct when generating the *ap^{R2}* landing site. It is *ap^{N5}* (isolation number c102.1). It contains a 19-bp deletion created at the site of the distal gRNA cutting site. The sequence flanking the deletion (indicated by -[]-) is as follows: TTTCGTGTGATTTCGG

GACC-[]-GTTCTGATCTCGGCTACAAA. Note that this allele is not associated with flanking attL and FRT sites.

Re-entry plasmid pDB345 consists of the following parts. The backbone is pUC57-KanR. In contains an attB site (54 bp) and a 35-bp polylinker consisting of the following unique restriction sites: HindIII_PacI_NheI_XhoI_MluI_SpeI. A 48-bp extended FRT site precedes *mini-yellow*, which serves as selectable marker. It is a 2650-bp KpnI/ClaI fragment mainly consisting of a yellow cDNA devoid of all known *yellow* enhancers. When inserted in the *apterous* locus, *mini-yellow* is expressed mainly in the wings of transgenic flies. However, its presence interferes with apterous function. In hemizygous $ap^{R2-WT[y+]}/ap^{DG3}$ flies, 94% of the wings showed a phenotype, which was usually restricted to the posterior wing compartment. Frequently, an enlargement of the posterior compartment could be observed (see *Figure 1—figure supplement 3C'*). This phenotype is reminiscent of $ap^{blot}$ (*Whittle, 1979*). This allele carries a *blood* retrotransposon 140 bp proximal to the distal end of OR463 (*Bieli et al., 2015a*). It is conceivable that the presence of *blood* or the *mini-yellow* marker (including the pDB345 backbone) within 55 or 195 bp, respectively, of the crucial m3 region is responsible for the phenotypic effect. Thus, the *mini-yellow* marker was removed by Flp-treatment from all $ap^{R2}$ transgenic inserts presented in this study. At the end of this procedure, the foreign DNA flanking OR463 fragments are the 56-bp attL on the proximal and the 34-bp FRT on the distal side. Importantly, removal of the foreign DNA led to a dramatic increase in hemizygous $ap^{R2-WT}/ap^{DG3}$ flies with normal wings. Most homozygous $ap^{R2-WT}$ flies developed normal wings. Few of them can have weak phenotypes like small notches and other weak margin defects or blisters (see arrow in *Figure 1—figure supplement 3D'*). In hemizygous $ap^{R2-WT}/ap^{DG3}$ flies, 95% wings are normal. The remaining wings can also show the enlarged posterior compartment phenotype seen in hemizygous $ap^{R2-WT(y+)}$ flies. It is possible that the remaining attL and FRT sites impede with complete rescue of the wing phenotype. Nevertheless, wing phenotypes of $ap^{R2}$ and $ap^{R2-WT}$ warrant a faithful dissection of the OR463 enhancer.

From a practical point of few, it might be of interest with what efficiency all the m3.1 substitution library transgenes were obtained. The library contains 22 different plasmids (1 wild-type and 21 mutants; synthesis and cloning by Genewiz). After injection of the plasmid mix into *y w vas-integrase; $ap^{R2}$/CyO* embryos, survivors were reared to adulthood and individually crossed with y w partners. From 34/141 fertile G0 crosses, 248 y$^+$ transformants were selected. These were individually crossed with *y w; $ap^{DG3}$/CyO, Dfd-YFP* partners. After a few days, the y$^+$ candidates were sacrificed and screened by PCR and sequencing. PCR primers were A185 (5'-CACCTGACTCAATAGCAAGACT-3') and m2 (5'-CAGTGTTCGGGTAATCAGGTG-3'). The resulting 1227-bp product was sequenced with primer P41 (5'-CTAATGCAATCGACAAACCCC-3'). After screening 104 randomly chosen candidates, all 22 possible alleles were represented by at least one candidate.

## Generation of the UAS-dCas plasmid, OR463 gRNAs, and control gRNA

The ORF of FLAG-NLS-dCas9-NLS was amplified from the plasmid pWalium20-10X*UAS*-3XFLAG-dCas9-VPR (Addgene: #78897) (*Lin et al., 2015*) using the primers: 5'-*ATCGCGGCCGCCGCCACCATGGATTACAAG*-3' and 5'-*GTGGTACCTCACACCTTCCTCTTCTTCTTGG*-3'. The fragment was subsequently cloned into the pUAStattB plasmid using the restriction sites NotI and KpnI.

U6-OR463x4 was generated using established protocols using the pCFD5 vector (*Port and Bullock, 2016*). sgRNA spacer sequences were (in 5' to 3' order within the array): (1) 5'-*TCACTCCCAGGCTCTGCCGA*-3', (2) 5'-*TCCAGAGATCGAGCGTCAGT*-3', (3) 5'-*ATTTGTAGCCGAGATCAGAA*-3', and (4) 5'-*AAAATATATGAATCCATTGC*-3'. As control gRNA, we used a gRNA binding an unrelated sequence in the iab5 regulatory region of *AbdB*. The gRNA was cloned following published protocols (*Port and Bullock, 2016*). The spacer sequence was 5'-*GTTGAGTTAGAGATCCCAGCAGG*-3'.

The pCFD5-OR463.gRNAx4 plasmid was injected in the BL25709 to generate flies used in this study.

## Generation of gRNAs targeting the Antp locus

U6-Antp.gRNAx3 was generated using established protocols using the pCFD5 vector (*Port and Bullock, 2016*). sgRNA spacer sequences were (in 5' to 3' order within the array): (1) 5'-*CTAGCTCTAGAGTCTGGTAC*-3', (2) 5'-*ATGACGCTGCCCCATCACAT*-3', and (3) 5'-*GTACGAGTTGGTGAAGTACG*-3'. The pCFD5-*Antp*.gRNAx3 plasmid was injected in the BL25709 to generate flies used in this study.

## Generation of CON2 and CON5 fluorescent reporters

mKate:NLS and eGFP:NLS fragments were synthesized by Genewiz and cloned via AgeI/SacII into pAttB-C2-LacZ and pAttB -C5-LacZ (*Bieli et al., 2015b*), respectively. Thereby, the LacZ ORF was substituted by that of the fluorescent proteins. Then, the new plasmids (pDB139 and pDB138) were introduced into landing site *zh-86Fb* and balanced stocks were established.

## Generation of Antp MARCM clones

Eggs of the indicated age were given a heat-shock at 37°C for 1 hr in a water bath. After incubation, tubes were transferred to 25°C until dissection. The genotype of the flies was:

   *y, w, hs-FLP, tubGal4, UAS-GFP /+ ; ; Antp$^{NS+RC3}$, FRT82B/ ubi-GFP, FRT82B.*

## Statistical analysis

The statistical analyses performed in *Figure 2* were *t*-tests. Prism software was employed to analyze the data and generate the graphs. p-values are indicated the figure legend.

## X-Gal staining

Third-instar larvae were dissected in cold PBS by opening the most anterior third of their cuticle and removing the internal organs. The cuticle was then reverted to expose the wing discs to the exterior. The tissues were fixed by incubation at RT for 15 min in 1% glutaraldehyde (Fluka) (in PBS) and subsequently washed with 0.1% Tween 20 (Fluka) in PBS under rotation at RT. Samples were then incubated using staining solution (440 µl $H_2O$, 25 µl 200 mM $NaP_i$ (pH 7.2), 15 µl 5 M NaCl (in $H_2O$), 0.5 µl 1 M $MgCl_2$, 5 µl 333 mM $K_4[Fe^{II}(CN)_6]H_2O$, 5 µl 333 mM $K_3[Fe^{III}(CN)_6]H_2O$, 2.5 µl 10% Tween-20, 8 µl 5% X-Gal (AppliChem) in dimethylformamide) at 37°C in the dark for 90 min. Finally, samples were washed with 0.1% Tween 20 and with PBS before mounting in 80% glycerol on a microscope slide.

## Bioinformatic analysis of apE

Sequence conservation of the OR463 fragment within the *ap* upstream intergenic region was analyzed across different dipteran species using the 'Cons 124 Insects' multiple-alignment track of the *D. melanogaster* dm6 genome on the UCSC Genome Browser (*Kent et al., 2002*, https://genome.ucsc.edu). Conservation scores were obtained from the phastCons (*Siepel et al., 2005*) and used to delineate conserved and less conserved blocks within OR463. Conserved transcription-factor-binding sites were predicted with MotEvo (*Arnold et al., 2012*), which defined four conserved modules (m1–m4) and six inter-modules (N1–N6). Additional motif analysis was performed using the JASPAR CORE Insecta database and the Target Explorer tool to cross-validate conserved binding site predictions and refine motif assignments within the enhancer.

# Acknowledgements

We thank Dr. Carlos Estella and Dr. Minkyoung Lee for their comments on the manuscript. We thank Basil Willi for the construction of the plasmid used for generating allele ap$^{R2.\Delta m3.5}$ and to Prof. Kristen Guss, Prof. Isabel Guerrero, and Dr. Natalia Azpiazu for the provided reagents. We are grateful to Karin Mauro, Bernadette Bruno, Gina Evora, Sermin Kösger, and Maria del Consuelo Zuluaga Gomez for constant and reliable supply of the world's best fly food. The work in the laboratory of M.A. was supported by grants from the Swiss National Science Foundation (310030_192659/1) and by funds from the Kanton Basel-Stadt and Basel-Land. GA was supported by 'Fellowships for Excellence' from the International PhD Program in Molecular Life Sciences of the Biozentrum, University of Basel.

# Additional information

## Competing interests

Dimitri Bieli: Affiliated with Mabylon AG; the author has no other competing interests to declare. Gordian Born: Affiliated with Arcondis; the author has no other competing interests to declare. The other authors declare that no competing interests exist.

## Funding

| Funder | Grant reference number | Author |
| --- | --- | --- |
| Swiss National Science Foundation | 310030_192659 | Markus Affolter |
| Swiss National Science Foundation | 310030B_176400 | Markus Affolter |

The funders had no role in study design, data collection, and interpretation, or the decision to submit the work for publication.

## Author contributions

Gustavo Aguilar, Conceptualization, Data curation, Formal analysis, Investigation, Writing – original draft, Project administration, Writing – review and editing; Michèle E Sickmann, Gordian Born, Conceptualization, Investigation; Dimitri Bieli, Conceptualization, Investigation, Writing – review and editing; Markus Affolter, Conceptualization, Funding acquisition, Writing – review and editing; Martin Müller, Conceptualization, Data curation, Investigation, Methodology, Writing – original draft, Writing – review and editing

## Author ORCIDs

Gustavo Aguilar ⬦ https://orcid.org/0000-0002-0524-9388
Markus Affolter ⬦ https://orcid.org/0000-0002-5171-0016
Martin Müller ⬦ https://orcid.org/0000-0002-0667-9574

Reviewer #1 (Public review): https://doi.org/10.7554/eLife.91713.4.sa1
Reviewer #3 (Public review): https://doi.org/10.7554/eLife.91713.4.sa2
Author response https://doi.org/10.7554/eLife.91713.4.sa3

# Additional files

## Supplementary files

Supplementary file 1. OR463 sequence and sub-regions. Conserved sub-regions are highlighted in red (m1–m4; most conserved), whereas less conserved subregions are highlighted in yellow (N1–N6). The chromosomal breakpoint of the *ap^Xasta* mutant and the insertion site of *ap^Blot* are also indicated.

Supplementary file 2. Fly stocks generated in this study.

Supplementary file 3. List of mutagenic primers employed for the generation of OR463 mutants.

MDAR checklist

## Data availability

All data generated or analyzed during this study are included in the manuscript and supporting files. All materials generated in this study are available from the corresponding author upon request.

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
