## [Editor Report · eLife Assessment]

This **important** paper presents the discovery of the molecular basis of differential apterous expression during early *Drosophila* wing disc development. The evidence supporting these conclusions is **compelling**, ranging from classical genetic approaches to state-of-the-art genetic engineering techniques. By opening new questions, this paper is expected to be of broad interest to developmental biologists and geneticists working on transcriptional regulation.

---

## [Referee Report · Reviewer #1 (Public review)]

Summary:

The *Drosophila* wing disc is an epithelial tissue which study has provided many insights into the genetic regulation of organ patterning and growth. One fundamental aspect of wing development is the positioning of the wing primordia, which occurs at the confluence of two developmental boundaries, the anterior-posterior and the dorsal-ventral. The dorsal-ventral boundary is determined by the domain of expression of the gene apterous, which is set early in the development of the wing disc. For this reason, the regulation of apterous expression is a fundamental aspect of wing formation.

In this manuscript the authors used state of the art genomic engineering and a bottom-up approach to analyze the contribution of a 463 base pair fragment of apterous regulatory DNA. They find compelling evidence about the inner structure of this regulatory DNA and the upstream transcription factors that likely bind to this DNA to regulate apterous early expression in the *Drosophila* wing disc.

Strengths:

This manuscript has several strengths concerning both the experimental techniques used to address a problem of gene regulation and the relevance of the subject. To identify the mode of operation of the 463 bp enhancer, the authors use a balanced combination of different experimental approaches. First, they use bioinformatic analysis (sequence conservation and identification of transcription factors binding sites) to identify individual modules within the 463 bp enhancer. Second, they identify the functional modules through genetic analysis by generating *Drosophila* strains with individual deletions. Each deletion is characterized by looking at the resulting adult phenotype and also by monitoring apterous expression in the mutant wing discs. They then use a clever method to interfere in a more dynamic manner with the function of the enhancer, by directing the expression of catalytically inactive Cas9 to specific regions of this DNA. Finally, they recur to a more classical genetic approach to uncover the relevance of candidate transcription factors, some of them previously known and others suggested by the bioinformatic analysis of the 463 bp sequence. This workflow is clearly reflected in the manuscript, and constitutes a great example of how to proceed experimentally in the analysis of regulatory DNA.

Weaknesses:

The previously pointed weakness (vg expression, P compartment specific effects, early vs late analysis of ap expression in mutants) has been thoroughly and satisfactorily addressed by the authors.

---

## [Referee Report · Reviewer #3 (Public review)]

In this manuscript, authors use the *Drosophila* wing as model system and combine state-of-the-art genetic engineering to identify and validate the molecular players mediating the activity of one of the cis-regulatory enhancers of the apterous gene involved in the regulation of its expression domain in the dorsal compartment of the wing primordium during larval development. The paper is subdivided into the following chapters/figures:

(1) In the first couple of figures, authors describe the methodology to genetically manipulate the apE enhancer (a cartoon summarizing all the previous work with this enhancer might help) and identify two well-conserved domains in the OR463 enhancer required for wing development (the m3 region whose deletion phenocopies OR463 deletion: loss of wing, and the m1 region, whose deletion gives rise to AP identify changes in the P compartment).

(2) In the following three figures, authors characterize the m1 regulatory region, identify HOX and ETS binding sites, functionally validate their role in wing development and the activity of the genes/proteins regulating their activity (eg-. Hth and Pointed) by their ability to phenocopy (when depleted) the m1 loss of function wing phenotype. Authors conclude that Hth and Pointed regulate apterous expression through the m1 region.

(3) In the last few figures, the authors perform similar experiments with the m3 regulatory region to conclude that the Grn and Antennapedia regulate apterous expression through the m3 enhancer.

Comments on revised version:

The authors have adequately addressed my major concerns.

---

## [Author Response]

The following is the authors’ response to the previous reviews

**Public Reviews:**

**Reviewer #1 (Public review):**
Summary:The *Drosophila* wing disc is an epithelial tissue which study has provided many insights into the genetic regulation of organ patterning and growth. One fundamental aspect of wing development is the positioning of the wing primordia, which occurs at the confluence of two developmental boundaries, the anterior-posterior and the dorsal-ventral. The dorsal-ventral boundary is determined by the domain of expression of the gene apterous, which is set early in the development of the wing disc. For this reason, the regulation of apterous expression is a fundamental aspect of wing formation.In this manuscript the authors used state of the art genomic engineering and a bottom-up approach to analyze the contribution of a 463 base pair fragment of apterous regulatory DNA. They find compelling evidence about the inner structure of this regulatory DNA and the upstream transcription factors that likely bind to this DNA to regulate apterous early expression in the *Drosophila* wing disc.Strengths:This manuscript has several strengths concerning both the experimental techniques used to address a problem of gene regulation and the relevance of the subject. To identify the mode of operation of the 463 bp enhancer, the authors use a balanced combination of different experimental approaches. First, they use bioinformatic analysis (sequence conservation and identification of transcription factors binding sites) to identify individual modules within the 463 bp enhancer. Second, they identify the functional modules through genetic analysis by generating *Drosophila* strains with individual deletions. Each deletion is characterized by looking at the resulting adult phenotype and also by monitoring apterous expression in the mutant wing discs. They then use a clever method to interfere in a more dynamic manner with the function of the enhancer, by directing the expression of catalytically inactive Cas9 to specific regions of this DNA. Finally, they recur to a more classical genetic approach to uncover the relevance of candidate transcription factors, some of them previously know and other suggested by the bioinformatic analysis of the 463 bp sequence. This workflow is clearly reflected in the manuscript, and constitute a great example of how to proceed experimentally in the analysis of regulatory DNA.Weaknesses:The previously pointed weakness (vg expression, P compartment specific effects, early vs late analysis of ap expression in mutants) have been throughly and satisfactorily addressed by the authors.

We thank the reviewer for the positive assessment of our manuscript as well as for the many constructive comments during its revision.

**Reviewer #3 (Public review):**
In this manuscript, authors use the *Drosophila* wing as model system and combine state-of-the-arte genetic engineering to identify and validate the molecular players mediating the activity of one of the cisregulatory enhancers of the apterous gene involved in the regulation of its expression domain in the dorsal compartment of the wing primordium during larval development. The paper is subdivided into the following chapters/figures:(1) In the first couple of figures, authors describe the methodology to genetically manipulate the apE enhancer (a cartoon summarizing all the previous work with this enhancer might help) and identify two well-conserved domains in the OR463 enhancer required for wing development (the m3 region whose deletion phenocopies OR463 deletion: loss of wing, and the m1 region, whose deletion gives rise to AP identify changes in the P compartment).(2) In the following three figures, authors characterize the m1 regulatory region, identify HOX and ETS binding sites, functionally validate their role in wing development and the activity of the genes/proteins regulating their activity (eg-. Hth and Pointed) by their ability to phenocopy (when depleted) the m1 loss of function wing phenotype. Authors conclude that Hth and Pointed regulate apterous expression through the m1 region.(3) In the last few figures, authors perform similar experiments with the m3 regulatory region to conclude that the Grn and Antennapedia regulate apterous expression through the m3 enhancer.My comments:Technically sound: As stated in my previous review, the work is technically excellent (authors use stateof-the-art genetic engineering to manipulate the enhancer and combine it with genetic analysis through RNAi and CRISPR/Cas9 and phenotypic characterization to functionally validate their findings), figures are nicely done and cartoons are self-explanatory.

We thank the reviewer for these positive comments.

Poor paper writing: The paper is too long and difficult to read/understand, many grammatical mistakes are found, and formatting is in some cases heterodox.

We thank the reviewer for this assessment. We have carefully revised the manuscript to improve clarity, readability, and consistency throughout. Specifically:

(1) Streamlined several sections to improve narrative flow. Specially in the abstract, model and dCas9 sections.

(2) Corrected grammatical issues across the manuscript. As the reviewer pointed out, we found many in the text. We are grateful the reviewer was insistent in this point.

(3) harmonized formatting and terminology. Many small inconsistencies were found in the figure legends, that have been largely adapted.

We believe these changes substantially improve the accessibility and overall presentation of the work. However, we have not shortened the manuscript, as we want to transmit the complexity of attempting to dissect non-coding regions, as well as not oversimplify the phenotypes obtained.

Science:(1) The question of "who is locating the relative position of the AP and DV boundaries in the developing wing?" is not resolved. I would then change the intro or reduce the tone of this question. Having said that, I agree that these results shed light on the wing phenotypes of some apterous alleles related to AP identify and growth and, as such, I congratulate the authors.

We appreciate this important point. We agree that our study does not fully resolve the upstream mechanisms that ultimately position the AP and DV boundaries. Our goal was instead to determine how the ap early enhancer (apE) contributes to the correct spatial relationship between these boundaries. To address the reviewer’s concern, we have revised the Introduction and Discussion to soften the framing of this question and to more clearly state the scope of our conclusions. We now emphasize that our work provides mechanistic insight into how apE function impacts DV/AP boundary organization, rather than claiming to fully resolve the upstream positioning mechanism.

(2) Identification of two TFs (Grain and Antp) mediating the regulation of apterous expression is interesting but some contextualization might be required. Data on Antp is not as convincing as data on Grn. I wonder whether Antp data can be removed at all.

We thank the reviewer for this thoughtful evaluation. We agree that the genetic evidence for Grain (Grn) is stronger and more direct than for Antennapedia (Antp). In response, we have revised the manuscript to more carefully calibrate the strength of our conclusions regarding Antp.

Specifically, we have:

Softened the language throughout to describe Antp as a candidate HOX input,

Explicitly stated that direct binding to the m3 site remains to be demonstrated biochemically, and

Clarified in the Discussion that our data support an early contributory role for Antp rather than establishing it as the definitive HOX factor acting at apE.

We believe retaining the Antp data is important because:

(1) The m3 site shows strong HOX dependency in vivo,

(2) Early Antp depletion produces clear defects in ap expression, and

(3) Recent literature supports an early requirement for Antp in wing development.

Together, these observations provide a coherent working model while appropriately acknowledging current limitations. We hope the reviewer agrees that the revised framing now appropriately reflects the strength of the evidence.

(3) I am not sure whether the term hemizygous is used properly

We use the term hemizygous as in classical genetics, in which an individual carrying an allele opposite a chromosomal deletion is considered hemizygous at that locus (see for example the entry for *ap^4^* mutant in the red book Lindsley and Zimm, The Genome of *Drosophila melanogaster*):

“… *ap4 /Df(2L) M4IA-54* hemizygote has nearly normal complement of bristles but otherwise resembles *ap4* homozygote (Butterworth and King, 1965).”